# Contemporary Approaches to Immunotherapy of Solid Tumors

**DOI:** 10.3390/cancers16122270

**Published:** 2024-06-19

**Authors:** Alla V. Kuznetsova, Xenia A. Glukhova, Olga P. Popova, Igor P. Beletsky, Alexey A. Ivanov

**Affiliations:** 1Laboratory of Molecular and Cellular Pathology, Russian University of Medicine (Formerly A.I. Evdokimov Moscow State University of Medicine and Dentistry), Ministry of Health of the Russian Federation, Bld 4, Dolgorukovskaya Str, 1127006 Moscow, Russia; avkuzn@list.ru (A.V.K.); petrovnapopova@rambler.ru (O.P.P.); 2Koltzov Institute of Developmental Biology, Russian Academy of Sciences, 26 Vavilov Street, 119334 Moscow, Russia; 3Onni Biotechnologies Ltd., Aalto University Campus, Metallimiehenkuja 10, 02150 Espoo, Finland; gkseniya@gmail.com (X.A.G.); ipbeletsky@gmail.com (I.P.B.)

**Keywords:** adoptive cell transfer, CAR-T cells, CAR-NK cells, CAR-macrophages, antibody-drug conjugate, immune checkpoint inhibitor, immune cell enhancers, tumor microenvironment

## Abstract

**Simple Summary:**

The last decade has seen dramatic changes in cancer treatment. Various methods of the immunotherapy of solid tumors, including treatment with targeted monoclonal antibodies, such as immune checkpoint inhibitors, bi- or trispecific immune cell enhancers, and antibody derivatives (e.g., antibody–drug conjugates), in addition to adoptive cell transfer or cell therapy, are being actively developed. A number of these approaches target tumor cells directly, while several other strategies aim to neutralize immunosuppressive elements in the tumor microenvironment (TME) and transform TME from immunologically “cold” to “hot”. Although many of these approaches have received clinical approval, each has inherent limitations that are considered in this review. Additionally, we discuss recent innovations in immunotherapy to improve clinical efficacy in solid tumors and strategies to overcome the limitations of various immunotherapy.

**Abstract:**

In recent years, the arrival of the immunotherapy industry has introduced the possibility of providing transformative, durable, and potentially curative outcomes for various forms of malignancies. However, further research has shown that there are a number of issues that significantly reduce the effectiveness of immunotherapy, especially in solid tumors. First of all, these problems are related to the protective mechanisms of the tumor and its microenvironment. Currently, major efforts are focused on overcoming protective mechanisms by using different adoptive cell therapy variants and modifications of genetically engineered constructs. In addition, a complex workforce is required to develop and implement these treatments. To overcome these significant challenges, innovative strategies and approaches are necessary to engineer more powerful variations of immunotherapy with improved antitumor activity and decreased toxicity. In this review, we discuss recent innovations in immunotherapy aimed at improving clinical efficacy in solid tumors, as well as strategies to overcome the limitations of various immunotherapies.

## 1. Introduction

The last decade has seen dramatic changes in cancer treatment with the active development of immunotherapy drugs that boost the immune response to eliminate cancer cells in infiltrated tissues [1,2] and drugs that target the tumor microenvironment (TME)—namely, cancer-associated fibroblasts (CAFs) and the extracellular matrix (ECM) [3,4,5].

The TME is now regarded as the master structure responsible for the tumor-mediated immunosuppressive network, making it the subject of intense research in the search for new therapeutic targets [3]. TME is composed of a heterogeneous population of non-malignant/normal cells, along which stromal cells (resident fibroblasts, CAFs, myofibroblasts, and pericytes), endothelial cells, neurons, tumor-infiltrating lymphocytes (TILs) such as B and T cells, natural killer cells (NK cells), and other cell types such as tumor-associated macrophages (TAMs), dendritic cells (DCs), and tumor-associated neutrophils (TANs). Additionally, the TME includes blood and lymphatic vessels, different components of the ECM, and signalling molecules [6,7,8]. Together, these components constitute TME, which is responsible for intercellular communication, tumor cell nutrition, signal transduction, and cancer cell behaviour [9].

Desmoplastic reaction is typical of solid tumors and is most prominent in pancreatic ductal adenocarcinoma (PDAC) [3,10]. The dense stroma contributes to a pro-tumorigenic environment, providing structural support for tumor cells and promoting angiogenesis; additionally, it suppresses immune cell infiltration, preventing the destruction of cancer cells and making them more resistant to chemotherapy, radiation therapy, and immunotherapy [11,12,13].

Immune cells infiltrating the TME play a pivotal role in tumor cell invasion and tumor progression (Figure 1), and immunological expression features directly correlate with the rate of tumor growth [14,15,16,17,18]. According to recent studies, T-cell dysfunction or T-cell “exhaustion”, which develops in response to persistent exposure to antigen, is a critical barrier to successful anti-tumor immunity. The dysfunctional or “exhausted” T cells present in malignant tumors are featured by the sustained and diverse expression of inhibitory receptors, called immune checkpoints, which are responsible for suppressing proliferation and pro-inflammatory cytokine production in cytotoxic lymphocytes (CTLs), the central effectors of anti-tumor immunity [19,20]. In addition, it has been shown that, in solid tumors, infiltrated NK cells respond to conditions inside the tumor and switch to a tumor-retained dormant state different from that of circulating NK cells in normal tissues. Adaptation to TME is accompanied by an impairment of the main effector functions of NKs—namely, the effective killing of cancer cells and the recruitment and activation of DCs [21]. DCs, at least some of them, instead of being directed to the lymph nodes, are trapped inside the tumor, where they become “exhausted”, with an impaired capacity to stimulate anti-tumor immune responses and the upregulation of signals that may even downregulate the anti-tumor function of other immune cells [22]. In essence, the prognosis of overall survival and the patient response to treatment are directly dependent on the immune landscape of the cancer [23,24]. This review focuses on the challenges and promising strategies for the immunotherapy of solid tumors.

## 2. Subtypes of Solid Tumors

Solid tumors are phenotypically diverse. They include many subtypes with different histological features, modes of dissemination, therapeutic responses, imaging features, prognostic information, and patient outcomes. Nevertheless, treatment selection based on these characteristics remains a great challenge [25,26,27].

The determination of molecular subtypes of different cancers has become a useful and promising addition to the analysis of morphological types to substantiate prognosis and patient response to treatment. A number of solid tumors have been classified using gene expression profiles [26,28,29,30]. Studies have shown that the molecular subtyping of malignancies reflects cardinal differences in clinical manifestations such as morbidity, response to treatment, disease progression, and survival. At the same time, the transcriptome subtyping has not yet brought it to the refinement of histopathological classifiers of tumor stages (classic anatomic tumor, nodules, and metastasis (TNM) classification) and better therapies suitable for a specific group of patients, i.e., therapies that target specific mechanisms of carcinogenesis. The issue of cellularity, TME heterogeneity, and the potential presence of normal tissue in transcriptomic studies are likely barriers to obtaining an optimal single classification that addresses all requirements. Despite significant genome sequencing efforts, without information on the distribution of immune cells in the TME, it is likely impossible to predict tumor/stromal features that are clearly relevant to immune interactions. Furthermore, the use of only one bioinformatics analysis without an immunohistochemical study to divide a cohort of patients into subtypes according to the degree of immune enrichment and stromal activation also does not clarify this problem [30,31,32,33]. At the same time, meta-analyses of published experimental data with immunohistochemical characterisation of TAMs have shown that the response to treatment and the overall survival of cancer patients depend on the density and polarity of TAMs, as well as the concomitant presence of TILs, granulocytic myeloid-derived suppressor cells (MDSCs), neutrophils, and other immune cells in TME, the levels of which are specific to different types of cancer [23,34,35,36,37,38]. A meta-analysis of published immunohistochemical data on NK cell infiltration to solid tumors showed that NK cell infiltration independently predicts the risk of death and overall survival. The prognostic value of NK is related to the tumor’s stage and grade, as well as its subtumoral location. Intraepithelial infiltration has been shown to be a better predictor of overall survival than NK infiltration in the tumor-adjacent stroma [24]. Thus, genome sequencing needs to be accompanied by analysis of the distribution of immune cells in TME; otherwise, it is impossible to correlate tumor/stroma features with immune interaction [32].

Combining gene expression assessment with immunohistochemistry and bulk tissue microarrays may be useful for classifying patients according to immune criteria, which can help to predict tumor outcome and to select specific treatments. For example, Knudsen et al., using a comprehensive analysis of genetic, stromal, and immunological features, described four subtypes of PDAC: cold, mutationally cold, hot, and mutationally active. The researchers showed that the immune infiltrate was diverse among PDAC cases, and enrichment with M2 macrophages (alternatively activated macrophages) and selected immune checkpoint regulators was specifically associated with survival [16]. The “cold” subtype was characterised by a low mutation burden and low levels of immune effector and suppressor cells. The “mutationally cold” subtype also contained a low number of mutations, but it showed a low stromal volume and an immature stromal type with high levels of monocarboxylate transporter 4 indicating a glycolytic and acidic microenvironment, with macrophages dominating the immune infiltrate. The “hot” and “mutationally active” subtypes exhibited a relatively high mutational burden, higher numbers of TILs, and peritumoral lymphocytes, as well as immune checkpoints (cytotoxic T lymphocyte-associated protein 4 (CTLA-4) programmed cell death ligand 1 (PD-L1)) and regulatory T cells (Treg) but exhibited different levels of TAM. The presence of macrophages (CD68-positive), and in particular, M2 macrophages (CD163-positive) had a negative impact on survival, whereas a subtype with low levels of neoantigens and minimal lymphocyte infiltration was associated with improved overall survival [16].

Three major immunophenotypes are distinguished according to the spatial distribution of CD8+ T lymphocytes in TME that correlate with the human response to anti-PD-L1/programmed cell death protein 1 (PD-1) therapy: the immune-desert, immune-excluded, and immune-inflamed phenotypes [39]. The first two subtypes are classified as cold tumors, while the last classifies hot tumors [40].

Three classes of TME are also distinguished according to the composition of the immune infiltrate and the nature of the inflammatory response: infiltrated-excluded (I-E), infiltrated-inflamed (I-I), and infiltrated-TLS (tertiary lymphoid structures). The last is a subclass of I-I TME and represents histological findings of TLS, lymphoid aggregates whose cellular composition is similar to that of lymph nodes. The I-I TMEs are considered immunologically “hot” tumors and are characterised by a high infiltration of CTLs expressing PD-1, as well as leukocytes and tumor cells expressing PD-L1 [5].

Tumors classified as I-E TME, which include PDAC, colorectal cancer, and melanoma, are weak immunogenic or “cold” tumors. The I-E TMEs are vastly infiltrated by immune cells but relatively depleted of CTLs in the tumor core. In I-E TME, these CTLs have a low expression of activation markers, granzyme B, and interferon gamma (IFN-γ), and are found at the tumor edges or “trapped” in so-called “fibrotic nests”. TAMs located also at the tumor periphery are thought to impede the entry of CTLs into the tumor core [5,41,42].

A tumor’s aberrant signalling pathways and oncogenes, driving the production of cytokines and chemokines that modulate the immune landscape, play a central role in the promotion of immunosuppressive TME. For instance, analysis of autochthonous mouse models of pancreatic and lung cancers with double mutations in KRAS/TP53 showed that co-mutations contribute to an increase in the number of cells expressing CD11b+, a marker of MDSCs and Treg. The infiltration of these two cell populations attenuates the response of CD4+ T helper type 1 (Th1) and CD8+ T cells, generating an immune tolerant environment [43]. In addition, oncogenic KRASG12D activation and p53 deletion promote increased expression of C-X-C chemokine receptor type 3 (CXCR3)/C-C chemokine receptor type 2 (CCR2)-associated chemokines and macrophage colony-stimulating factor (M-CSF), resulting in an increased number of CD11b+ TAMs with a putative immunosuppressive function [43,44]. Thus, the activation and infiltration of MDSCs, Treg, and TAM into the tumor, accompanied by the loss of CTLs activation signals and the exclusion of CTLs from the tumor core, are hallmarks of immunological ignorance in which the immune system fails to recognise and fight the malignancy. The situation is exacerbated by the low mutational load of the tumor and the enhancing immunosuppressiveness of the TME, which interferes with the priming, transport, and anti-cancer functioning of T- and NK-cells [5,45].

As can be seen above, there are different approaches to subtyping solid tumors, but they all boil down to the existence of “cold” and “hot” tumors, the differences between which are reflected in Figure 2.

Thus, certain immune populations may play an anti-tumor or pro-tumor role; the balance of their activity is determined by TME and may predict the response to treatment and overall survival. In turn, the tumor itself, through its underlying mutations, progression (stage and grade), vascularisation, and metabolism, as well as the soluble factors it produces, also contributes to TME by influencing immune cell infiltration and activation. Immune cells with pro-tumor activity infiltrating solid tumors should be considered as a therapeutic target. In general, interactions in TME are complex, and identifying key features of prognostic and therapeutic importance is crucial for the development of effective immunotherapy for solid tumors [24].

## 3. The Cancer Immunotherapy Strategies

Immunotherapeutic approaches encompass a range of therapies that focus on the patient’s immune system. These currently comprise therapy with immunomodulators, oncolytic viruses, and tumor-specific vaccines (mRNA vaccines), as well as treatment with targeted monoclonal antibodies (mAbs), including antibody-drug conjugates (ADCs), immune checkpoint inhibitors (ICIs) and bi- or trispecific immune cell enhancers (ICEs), and adoptive cell transfer (ACT) or cell therapy. A number of these approaches target tumor cells directly, while several other strategies aim to neutralise the immunosuppressive elements in TME and transform TME from immunologically “cold” to “hot”. This aim can be reached by enhancing the effector function of endogenous immune cells, improving the infiltration of CTLs, NK cells, and M1 macrophages, and eliminating or inhibiting the infiltration of Treg, MDSCs, and TAMs. Although many of these approaches have received clinical approval, each has inherent limitations that are considered below [40,45,46,47].

### 3.1. Monoclonal Antibodies (mAbs)

The use of mAbs has changed the paradigm of cancer therapy by precisely targeting tumor surface antigens, which, compared to chemotherapy, dramatically reduces off-target toxicity. In recent decades, a significant number of mAbs have been approved for the treatment of various solid tumors [48]. However, treatment with monoclonal antibodies alone is often insufficient due to persistent problems such as off-target toxicity, immunogenicity, resistance, and non-response in a significant proportion of patients. Hence, a new concept known as ADC has emerged, which was conceived to bridge the gap between mAbs and small-molecule cytotoxicity [49].

#### 3.1.1. Antibody-Drug Conjugate (ADC)

An ADC consists of a tumor-targeted mAb and a cytotoxic payload linked via a chemical linker. Currently, in addition to the 14 ADCs already approved by the U.S. Food and Drug Administration (FDA) for the treatment of malignant haematological and solid tumors, more than 100 ADC candidates, differing in their antigenic specificity (cancer indication), linker structure, cytotoxic compounds, and cleavage mechanism, are being evaluated in preclinical and extended clinical trials. An ideal ADC should remain stable in circulation, accurately reach the therapeutic target (target antigen), internalise upon binding to it, and release the cytotoxic agent within cancer cells; ultimately the cytotoxic agent, once in the cytoplasm, should cause cell death by affecting the DNA, DNA-servicing proteins, or cytoskeletal components (microtubules, actin, intermediate filaments and septins) [49,50,51,52]. When selecting a candidate target antigen for ADC, the following features of the target antigen should be considered: (1) its location, as the target antigen should be surface-exposed or extracellular rather than intracellular for better ADC recognition; (2) target antigen level of expression—it should be expressed exclusively or predominantly in target cancer cells but not in healthy cells to ensure the specificity of the therapeutic agent and avoid off-target toxicity; (3) antigen secretion—the target antigen should not be secreted because secreted antigen in the circulation may cause off-target activation of ADC, activation outside the tumor tissue, resulting in reduced tumor targeting and systemic toxicity; and (4) antigen internalisation—the target antigen should be ideally suited for internalisation of the construct: the antibody should induce rapid receptor internalisation, endosomal transport, and lysosomal processing [49,50] (Figure 3). If the ADC payload is sufficiently permeable for cytoplasmic membranes (lipophilic payloads), it can diffuse out of the cell in which it was released and kill surrounding cancer cells, which can be either positive or negative to the target antigen. This mechanism, the so-called “bystander killing” effect, not only enhances the cytotoxicity of ADC, but also allows it to target tumors with heterogeneous expression of the target antigen, thereby extending the therapeutic effect [52,53,54].

A number of target antigens have been identified for ADC use in solid tumors, including human epidermal growth factor receptor 2 (HER2), epidermal growth factor receptor (EGFR), carcinoembryonic antigen-related cell adhesion molecule 5 (CEACAM5), trophoblast cell surface antigen 2 (Trop-2), Claudin18, nectin-4/poliovirus receptor like 4 (PVRL4), and others [52].

For example, the following ADCs have been approved by the FDA for the treatment of breast cancer: Kadcyla^®^ (ado-trastuzumab emtansine) and Enhertu^®^ (fam-trastuzumab deruxtecan-nxki), target antigen for which is HER2; and Trodelvy^®^ (sacituzumab govitecan-hziy), the target antigen for which is Trop-2. Trop-2 is a transmembrane glycoprotein with low expression in normal, healthy tissues. Studies suggest that Trop-2 may play a role in tumor progression, given its involvement in several molecular pathways associated with cancer development. The high level of Trop-2 expression correlates with a poor prognosis in a number of cancers [55]. Sacituzumab govitecan contains the active metabolite SN-38 of the anticancer drug irinotecan that inhibits the enzyme nuclear topoisomerase I conjugated to a humanised antibody against Trop-2. When this ADC is internalised, its pH-sensitive linker ensures the release of SN-38 into the acidic environment of the tumor cell and TME, promoting a bystander effect on neighbouring cancer cells [56]. Patients with advanced/metastatic solid tumors including triple negative breast cancer, urothelial cancer, non-small cell lung cancer, small cell lung cancer, adenocarcinoma of endometrium, stomach, oesophagus, ovary, colorectal cancer, etc. are currently enrolled in a Phase I clinical trial to study the safety, tolerability and pharmacokinetics of the Trop-2-targeted drug FDA018-ADC (NCT05174637).

Another example is the cell adhesion molecule CEACAM5, a glycoprotein that is overexpressed in carcinomas of the gastrointestinal tract, genitourinary and respiratory systems, and breast cancer, and is usually not significantly internalised. A novel anti-CEACAM5 ADC, SAR408701, consisting of an anti-CEACAM5 antibody coupled to the microtubule-disrupting agent DM4 via a cleavable linker, has shown efficacy in preclinical studies [57,58]. SAR408701 is in Phase I clinical trials in patients with CEACAM5-positive tumors (NCT03324113, NCT02187848).

Challita-Eid et al. identified nectin-4 as a potential target antigen in various epithelial malignancies by means of suppressive subtractive hybridisation. Nectin-4 is a transmembrane protein that, together with cadherin, participates in the formation and maintenance of adhesion contacts. The researchers developed the ADC, enfortumab vedotin-ejfv/Padcev^®^, also known as ASG-22ME, comprising the anti-nectin-4 antibody linked to the microtubule-disrupting drug, monomethyl auristatin E. Treatment of human breast, bladder, pancreas and lung cancer xenografts in mouse models with enfortumab vedotin suppressed the growth of all four tumor types and ensured the regression of breast and bladder xenografts [59]. Enfortumab vedotin is currently FDA approved for the treatment of adult patients with urothelial cancer.

It is worth noting that third-generation ADCs have less toxicity and greater anticancer activity, as well as higher stability, compared to earlier ADCs [50]. However, the clinical development of ADCs for the treatment of solid tumors is hampered mainly by systemic toxicity due to off-target release of the payload, difficulties in penetrating deep into the tumor through the dense stroma, and the heterogeneous expression of target antigens. Therefore, in order to address these shortcomings and to increase the therapeutic index, more and more attention is now being paid to the development of non-internalising ADCs with extracellular payload release mechanisms [48,52,60,61]. This alternative approach avoids potentially inefficient internalisation, eliminates the dependence on high expression of target antigens, and broadens the spectrum of antigens from tumor cell antigens alone to targets in the TME, i.e., target proteins mediating intercellular and cell-matrix interactions, secreted proteins and proteins of the stroma and vessels (vasculature), components of the neovascular system, and subendothelial ECM. This approach is also referred to as “cancer stromal targeting therapy” [49,50,52]. ADCs targeting dense tumor stroma can potentially induce cancer cell death by reducing the concentration of growth factors produced by TME stromal cells. Since cancer cell survival depends on angiogenesis and matrix factors, ADCs may have higher efficacy by targeting such tissues. In addition, the genome of stromal cells in the TME is more stable than the genome of cancer cells, which reduces the likelihood of drug resistance caused by mutations [50,62,63]. In preclinical studies, non-internalising ADCs targeting TME’s proteins (Figure 3) such as galectin-3-binding protein, (Gal-3-BP, a.k.a. LGALS3BP, 90 K, Mac2-BP) [64,65], leucine-rich alpha-2-glycoprotein 1 (LRG1) [66], matrix metalloproteinase-9 (MMP-9) [67], collagen IV [68], fibronectin [69], tenascin-C [70], and fibrin [71] have shown to be effective. In addition, the non-internalising approach of ADC may be promising for targeting immune checkpoints, as their blockade reactivates the cytotoxicity of CD8+ T cells and enhances the antitumor immune response in the TME [72,73].

#### 3.1.2. The Immune Checkpoint Inhibitors (ICIs) in the Context of Cancer Treatment

There have been two important discoveries in the history of immune checkpoints. It has emerged that blockade of CTLA-4/CD80/CD86, the “brakes” of the immune system, and the blockade of the PD-1/PD-L1 axis, which suppresses T-lymphocyte proliferation, survival, and effector functions, can potentially reactivate “depleted” T cells in infectious diseases and cancer. This has led to the active development of ICIs, which “fuel” the immune response by activating intratumoral cytotoxic T lymphocytes and enhancing T cell infiltration in TME [74,75,76,77]. For example, anti-PD-1 drugs such as nivolumab (Opdivo^®^) or pembrolizumab (Keytruda^®^), anti-PD-L1 drugs, such as durvalumab (Imfinzi^®^) or atezolizumab (Tencentriq^®^), or anti-CTLA-4 drugs such as ipilimumab (Yervoy^®^) or tremelimumab (Imjudo^®^), are authorised for clinical use. These drugs have shown impressive results in several late stages of tumor progression, including metastatic melanoma, lung cancer, kidney cancer, bladder cancer, and head and neck cancer [78,79,80].

However, although ICIs can awaken immune system and restore “exhausted” T cells, this newfound anti-tumor state is often transient. A further problem with the use of ICIs is that not all patients respond to such treatment; some patients may initially respond but develop resistance over time, eventually leading to disease progression; response rates vary considerably by cancer type [47,78,80]. For example, ICI monotherapy or combination ICI therapy with standard chemotherapy, approaches that are effective in lung and breast cancer, have demonstrated only limited activity against PDAC and may not provide adequate response for patients with advanced pancreatic cancer. An exception in PDAC is the MMR-D patients (a rare subgroup with microsatellite instability or mismatch repair deficiency), in whom therapy with ICI has a beneficial effect [45,81,82]. To overcome resistance to immune checkpoint blockade in patients with high levels of infiltrated MDSCs in tumors IPI-549, a selective inhibitor of the gamma isoform of phosphoinositide 3-kinase gamma (PI3Kγ), which is highly expressed in myeloid cells, is being used [83]. IPI-549 is currently being evaluated in a Phase I clinical trial (NCT02637531).

Another approach is to use a combination of anti-PD-1/PD-L1 and anti-CTLA-4 mAbs to target non-redundant pathways. A randomised Phase II trial of nivolumab with ipilimumab in combination with stereotactic body radiotherapy for refractory metastatic pancreatic cancer provided durable clinical benefit in a small proportion of patients, including prolonged, sustained partial responses (NCT02866383). However, comparative trials of combination therapy with durvalumab (MEDI4736) and tremelimumab in patients with metastatic solid tumors have been discontinued due to non-encouraging results (NCT03982173). The addition of standard chemotherapy, which reduces tumor burden by targeting antigens and directly affecting the immunosuppressive compartment of TME, may be advantageous to enhance the anti-tumor effect of ICIs. Recently, patients with borderline resectable, locally advanced, or metastatic pancreatic cancer were recruited to study the safety and synergism of nivolumab and ipilimumab in combination with gemcitabine and nab-paclitaxel followed by immune-chemoradiation (NCT04247165). Another approach that is seen as potentially improving survival in patients with PDAC is combination therapy of ICI with inhibition of transforming growth factor beta (TGF-β) signaling. However, a clinical trial of a bifunctional fusion protein targeting TGF-β and PD-L1 (M7824) in combination with gemcitabine in adult patients with previously treated advanced pancreatic adenocarcinoma was halted after one treatment-related death (NCT03451773). When using ICIs, some patients experience immune-related adverse effects (irAE) due to over-activation of the immune system, such as colitis, hepatitis, and skin reactions. The type of side effect depends on the combination of drugs used, but the risk is much lower compared to other treatments such as chemotherapy or stem cell transplantation. Further research is obviously needed to better understand and treat these complications [80]. Different tumor types employ different strategies to overcome the immune response, resulting in a lack of ICIs efficacy. These include upregulation of immune checkpoint molecules by tumor cells and TME, downregulation of the human leukocyte antigen (HLA) molecules or impaired antigen presentation to antitumor T cells, and low tumor immunogenicity that prevents a cell-mediated antitumor response [84].

#### 3.1.3. Bi- and Trispecific Immune Cell Engagers (ICEs) for Immunotherapy of Solid Tumors

ICEs have been specifically designed to redirect immune cells to surface tumor-associated antigens (TAAs) for HLA-independent elimination of cancer cells and generation of immune responses against poorly immunogenic tumors. Most ICEs are trans-binding bispecific antibodies (bsAbs) consisting of two linked single-chain antibodies, one arm (scFv) of which binds TAAs and the other is directed to a cytotoxic trigger molecule, more precisely, to an activating receptor of effector cells including CD3 or costimulatory molecules such as CD28 or 4-1BB for recruiting T cells, CD16a (also known as FcγRIII) or NKG2D for NK cells, and CD64 (also known as FcγRI) for cytotoxic/phagocytic cells (Figure 4). More than 100 multispecific antibodies (bsAbs, trispecific antibodies (tsAbs) and even tetraspecific antibodies) are currently in clinical trials [84,85].

It is believed that bispecific T cell engagers (BiTEs) are significantly more effective than conventional mAbs at recognising the same tumor antigens. The formation of an “artificial” immunological synapse at the interface between target cells and T cells is accompanied by a redistribution of the signalling and secretory granules in T cells, leading to the release of cytokines and lytic granules with perforins and granzymes. Such contact-dependent cytotoxicity is probably the main mechanism of BiTE-induced direct killing of tumor cells. In addition, the antitumor activity of ICEs, as well as in ADC, may be promoted by the “bystander killing” effect [84,86,87]. Two BiTEs targeting CD3/CD20 (glofitamab-gxbm/Columvi and epcoritamab-bysp/Epkinly) have recently been approved for medical use in the United States, the European Union, and Canada. These bsAbs are indicated for the treatment of adults with relapsed or refractory diffuse large B-cell lymphoma [58]. Another example of a BiTE is elranatamab-bcmm/Elrexfio targeting CD3/B-cell maturation antigen (BCMA), a drug approved by the FDA for the treatment of multiple myeloma [88]. However, BiTEs have shown limited clinical efficacy in the treatment of solid tumors. This is partly due to the difficulty in penetrating tumor tissue and the lack of T cells infiltrating TME. In addition, the use of BiTEs for the treatment of solid tumors is accompanied by extratumoral toxicity. For example, when using solitomab, a BiTE construct targeting epithelial cell adhesion molecule (EpCAM) and CD3 (EpCAMxCD3), dose-limiting toxicities, including severe diarrhoea and elevated liver enzyme levels, were observed in 15 out of 65 patients with relapsed/refractory solid tumors, preventing dose escalation to potentially therapeutic levels [89,90].

A similar strategy was developed based on the BiTE concept to create NK cell engager (NKCE) bsAbs called bispecific killer cell engagers (BiKEs). An example of BiKEs is the bispecific anti-HER2xNKG2D antibodies that simultaneously interact with HER2 on tumor cells and NKG2D on NK cells. NKG2D, a member of the NKG2 family, is expressed as a homodimer together with the adaptor protein DAP10 (Figure 4); it is a key activating receptor for NK cells, and its expression is also found on T cell populations, including CD8+, NKT and γδ T cell subpopulations. HER2xNKG2D BiKE induces cytotoxicity of unstimulated NK cells in a tumor-specific manner, regardless of their apparent affinities and epitopes. As NKG2D can bind to eight types of ligands expressed in solid tumors, it potentially can be applied to a wider range of tumor cells [85,91,92,93].

In addition, as macrophages are a majority of the immune cell infiltrate in the TME of a solid tumor, the creation of bispecific macrophage engagers (BiMEs) is considered to be a promising strategy in the treatment of solid tumors. Encouraging results were obtained with early bsAbs targeting HER2 and CD64 (MDX-H210). These demonstrated high antibody-dependent cell-mediated cytotoxicity (ADCC) and efficient cytotoxic lysis of solid tumor cells in preclinical models, but showed limited efficacy in clinical trials [84,94,95,96]. In search of a suitable target for macrophage, various other receptors have been considered, including the transmembrane protein signal regulatory protein alpha (SIRPα/SHPS-1/CD172a). The CD47 protein expressed on the surface of various tumor cells, when interacting with SIRPα on macrophages, gives the latter a “don’t eat me” signal, thus inhibiting their phagocytic activity. Hence, by targeting both pathways, macrophage phagocytic activity can be restored and directed towards tumor cells. Several bsAbs have been developed to block the CD47/SIRPα interaction. For example, TQB2928 that mediates blockade of CD47 and SIRPα is in the first-in-human Phase I trial of TQB2928 in patients with advanced solid tumors (NCT04854681). Recently, a panel of bispecific TAA/SIRPα antibodies has been established as potential BiMEs. Claudin18.2, EGFR, PD-L1, and DLL-3 (where the last is a Notch ligand) were used as TAAs. BiME treatment showed high antitumor efficacy in several syngeneic tumor models in mice. Interestingly, BiMEs also activated myeloid and T cells [97,98,99].

Advances in antibody engineering, which enable the large-scale production of antibody constructs and the creation of antibody molecules with artificial structures, new specificity, and effector functions, allow the development of new tsAbs, which appear promising for the treatment of solid tumors. The presence of a third binding motif in tsAb proteins should contribute by either recruiting an additional TAA to augment specificity and avoid immune escape or by the costimulating of tumor-associated immune cells. Several trispecific T-cell engagers (TriTEs) and trispecific NK-cell engagers (TriKEs) are currently known.

Of the recent developments of TriTEs, a promising study showing that a tsAb to HER2/CD3/CD28 promotes breast cancer cell regression in a humanised mouse model is noteworthy. The anti-cancer effect was found to be mediated by CD4 T cells through the inhibition of tumor cell cycle progression as well as enhancement of pro-inflammatory signalling [100]. Another approach tested on human liver cancer in NOD/SCID mice with patient-derived xenografts is based on the use of trispecific nanobodies (humanised camelid heavy chain antibodies without light chains) targeting CD3ε on T cells, fibroblast activation protein (FAP) on CAFs, and the immune checkpoint protein PD-1. These nanobody-TriTEs (nb-TriTEs) have been shown to suppress tumor growth, improve survival, and increase T-cell infiltration in TME [101,102]. Next, it is worth noting the interesting TriTE construction specific for CD3/EGFR/EpCAM. It provided specific cytolysis of EGFR- and/or EpCAM-expressing colorectal cancer cells and significantly prolonged the survival of tumor-bearing mice. However, the cytotoxicity and IFN-γ secretion stimulated by the corresponding bispecific light T-cell engagers used as controls were found to be markedly higher than that of TriTEs in the case of tumor cells expressing only one of the target antigens. A similar observation was made in the aforementioned work with nanobodies. Nb-TriTEs enhanced the lysis of FAP+ target cells, whereas an enhanced killing effect was not observed against FAP-negative target cells [102,103]. This indicates that increasing the specificity of engagers leads to a narrowing of the cell population in tumor to be treated. This is a serious limitation for the therapy of antigenically heterogeneous tumors.

To enhance the therapeutic effect of NK cells, tri- and even tetraspecific constructs targeting more tumor antigens or acting on additional activating receptors have been created. An interesting example of TriKE targeting a solid tumor is a trispecific molecule that provides enhanced binding to CD16/NKp46/EGFR. This molecule has been shown to promote efficient lysis of human lung carcinoma cells [104]. TriKE’s best-known developments were created by researchers from the University of Minnesota (USA) based on an original technology platform that provides targeting of CD16 on NK cells and TAA on tumors and includes an IL-15 linker between them, capable of stimulating NK cell proliferation and expansion. They have developed constructs in which the TAA can be EpCAM, CD33, B7-H3 (B7 Homolog 3/CD276, B7 family immunoregulatory protein), HER2, CLEC12A (C-type lectin domain family 12 member A), TEM8 (tumor endothelial marker 8), or mesothelin [105,106,107,108,109,110,111]. This team was able to create a tetraspecific construct (tetraspecific NK-cell engager, TetraKE) on the same technology platform where the TAAs were EpCAM and CD133. This TetraKE was reported to promote proliferation and increase the survival and killing activity of NK cells [112]. It should be noted that, as with TriTEs, the TriKEs exclusively target the tumor carrying the specific TAA; although, one would think that both IL-15 and the targeting CD16 would be sufficient to enhance NK activity, albeit to a lesser extent than TriKE. In addition, the lack of cytotoxic activity of NK cells isolated from healthy Minnesota donors is noteworthy. For instance, unlike NK cells isolated from donors from other geographic regions, Minnesota cells do not kill HT29 tumor cells without BiKE or TriKE [113,114]. TriKE drugs developed by the University of Minnesota team to target solid tumors (GTB-4550, GTB-5550, GTB-6550, where TAAs are PD-L1, B7-H3 and HER2, respectively) are currently in preclinical studies (sponsored by GT Biopharma, Inc.) [115,116]. In parallel, Dragonfly Therapeutics, Inc. is presenting, in preclinical and clinical trials, at least 15 drugs known as tri-specific NK engager therapy (TriNKET) targeting two activating NK cell receptors, CD16a and NKG2D, and TAA. Some of them target solid tumors carrying HER2 or EGFR (DF1001 (NCT04143711) and DF9001 (NCT05597839)), respectively [117,118,119]. The advantage of BiKEs and TriKEs is considered to be increased biodistribution, due to their low molecular weight and low immunogenicity. However, it is still questionable whether the increased biodistribution will cause the additional infiltration of NK cells into the TME or whether it will only affect already infiltrated NK cells. Since BiKEs and TriKEs are fully artificial recombinant 50–75 kDa proteins foreign to humans, their low immunogenicity compared to therapeutic antibodies remains an open question. Ongoing trials will provide reliable answers to these questions.

### 3.2. The Cellular Therapy/The Adoptive Cell Transfer (ACT)

#### 3.2.1. T-Cell-Based Therapy

T-cell approaches of ACT include (I) cancer-specific T-cells such as TILs that are isolated from a patient’s tumor, propagated ex vivo and reintroduced to the patient; and (II) endogenous T-cells derived from peripheral blood that are then exogenously primed with specific peptide/cancer antigen or genetically modified. In turn, genetically modified T cells are distinguished into (a) T-cell receptor (TCR) T-cells that recognise TAA specific for a given tumor, (b) chimeric antigen receptor (CAR)-T cells that recognise TAA specific for some type of tumor, and (c) T cells secreting T cell-engager antibodies (CART.BiTE; STAb-T cells) [78,120,121,122].

##### Tumor-Infiltrating Lymphocyte (TIL) Therapy

The development of TIL therapy as the first antigen-specific T-cell therapy against melanoma began almost 40 years ago [123]. On 16 February 2024, the FDA approved lifileucel (Amtagvi), the first skin melanoma treatment that uses TILs. Despite the clinical benefits of TIL therapy, there are many challenges associated with it. TIL therapy involves precise surgical removal of tumor tissue, isolation, and culturing of TILs. The isolation, analysis, and culturing of TILs requires highly trained and skilled researchers and clinicians as well as superior biomedical techniques and equipment. Therefore, only a few medical centres are capable of providing this therapy. “The relatively high doses of … chemotherapy given before treatment and the high dose Interleukin-2 after [lifileucel] infusion currently limit the types of people who can undergo it,” Alexander Shoushtari, M.D., of Memorial Sloan Kettering Cancer Center, reports [124]. In addition, tumor heterogeneity and the variable presence of TILs in different tumors affect the success of TIL therapy [47,125].

##### TCR T-Cell Therapy

To overcome the limitations of TIL, unmodified peptide-stimulated T cells and genetically modified TCR T cells have been tested in clinical trials with promising results. TCR T-cell therapy represents an outstanding alternative with a number of advantages. Unlike CARs, TCRs can recognise antigens derived from membrane and intracellular proteins and represented in the HLA system, whereas CAR-T cells only recognise cell surface antigens. However, TCR T-cell therapy is restricted to the HLA presenting antigen allele, which limits the patient population that can be subjected to such TCR-T-cell therapy. It also means that its efficacy may be affected by changes in HLA expression or antigen processing in cancer cells. In addition, the density of antigens required to activate TCR T-cells is from one per cell compared to 200 for CAR-T cells. This higher sensitivity may improve the targeting and elimination of tumor cells. Also, some authors believe that the high avidity of TCR T-cells and the lower affinity of TCR to target compared to CAR allow each TCR-T cell to “scan” and destroy multiple antigen-presenting tumor cells [47,126,127,128,129]. On 31 January 2024, the biologics license application for the first TCR T-cell therapy for the treatment of synovial sarcoma (afami-cel, formerly ADP-A2M4) was accepted by the FDA with priority review.

##### CAR-T Cell Therapy

CAR-T cells represent the most clinically advanced ACT technology. Depending on the origin of the T cells, a distinction is made between autologous (from the patient’s blood) and allogeneic (HLA matched, from the blood of a healthy donor) CAR-T-cell therapies. CAR is a hybrid receptor composed of three structural domains: an ectodomain containing a tumor antigen recognition domain known as scFv; a hinge region (or spacer), a hydrophobic transmembrane domain spanning the cell membrane that typically originates from CD8α or CD28; and an endodomain, also known as an intracellular signalling domain or cytoplasmic tail (Figure 5A). CARs direct T cells to predetermined antigens. The main advantage of CAR-T cells over other forms of adoptive T-cell therapy is that they can recognise cancer antigens in the absence of MHC presentation (Figure 6). Moreover, the “live drug” nature of CAR-T cells, which allows them to proliferate and persist in the patient, although raising safety concerns, provides a sustained antitumor response that many other therapies, such as mAbs and bi- and trispecific ICEs, cannot achieve [47,120,130].

From its discovery to the present, the CAR structure has endured significant changes, from its first to its fifth generation (Figure 5B), offering new therapeutic alternatives for cancer patients. In an attempt to enhance activation, resistance, proliferation, safety, and efficacy, the CAR structure has been redesigned by modifying the endodomain and adding costimulatory molecules. First-generation CAR-T cells contain only immunoreceptor tyrosine-based activation motif (ITAM) in the intracellular domain, which, in most cases, is the CD3ζ chain. Second-generation CAR-T cells include a single costimulatory molecule (i.e., CD28 or 4-1BB/CD137). Currently, all FDA-approved CAR-T therapies are based on second-generation constructs. Third-generation CAR-T cells comprise two costimulatory molecules in tandem (i.e., CD28 in combination with 4-1BB). Fourth-generation CAR-T cells are based on second-generation CARs paired with the retroviral expression cassette for cytokines. These T cells are also referred to as T cells redirected for universal cytokine-mediated killing (TRUCK). Fifth-generation CAR-T cells or “next-generation” CAR-T cells are also based on second-generation CARs with the addition of truncated cytoplasmic domain of IL-2 receptor β (∆IL-2Rβ) with a binding site for transcription factors such as STAT3/5 allowing for JAK/STAT pathway activation. However, this fifth generation still needs to be extensively tested, as it is well known that the over-activation of STAT3 leads to immunosuppression and transformation, and the activation of JAK/STAT pathways is widespread in various T-cell malignancies [47,120,131,132]. Many aspects of CAR design, such as epitope selection, affinity, hinge region length, and the comparison of different combinations of intracellular signalling domains, are still under active examination.

Since scFvs have noticeable limitations such as stimulation of self-aggregation of the CAR molecule, which, in turn, can lead to premature CAR activation and exhaustion of the CAR-T cell effector potential, alternative binding domains have been explored. These include nanobodies, recombinant antigen-specific scFvs derived from the heavy chain (VHH) of mAb; designed ankyrin repeat proteins (DARPins); and the use of natural receptors and ligands for antigen targeting [133,134].

It should be mentioned that the scFv of many CARs currently in clinical development are derived from murine or other non-human mAbs, and a number of patients with solid tumors have been found to have antibodies to compounds that do not normally exist in humans, so-called anti-idiotypic antibodies. This suggests that CAR-T cells are capable of inducing a humoral immune response against CAR, its foreign components, or residual proteins originating from gene transfer vectors that are inherently immunogenic. This demonstrates the need to minimise the inclusion of non-human components in CAR-T cell constructs for the treatment of solid tumors [135].

A wide spectrum of antigens overexpressed in haematological malignancies and solid tumors has been investigated in preclinical models and clinical trials with CAR-T cells. For example, of the 19 target antigens for CAR-T cells that have been well studied in preclinical studies in breast cancer, 12 are in clinical trials [136]. Overall, 22 target antigens have been described as investigational in patients with solid tumors in ongoing clinical trials worldwide [137]. However, to date, the FDA has only approved CAR-T-cell therapies targeting one of two B-cell antigens: CD19 (axicabtagene ciloleucel/Yescarta, tisagenlecleucel/Kymriah, brexucabtagene autoleucel/Tecartus, and lisocabtagene maraleucel/Breyanzi) and BCMA (idecabtagene vicleucel/Abecma and ciltacabtagene autoleucel/Carvykti), which are intended as second and next-line systemic therapies for certain subgroups of B-cell leukaemia or lymphoma. Comparable efficacy of CAR-T-cell therapy in solid tumors and more haematological malignancies has not yet been achieved [136,137,138,139].

Clinical trials of first-generation CAR-T cells directed against carbonic anhydrase IX (CAIX), neuronal cell adhesion molecule L1 (NCAM-L1)/CD171, folate receptor alpha (FR-α), GD2, and second- or third-generation CAR-T cells, directed against HER2, although they have demonstrated feasibility, have not provided a significant benefit to patients, showing limited activity and frequent toxicity, and have been generally disappointing [140]. These results may be due to several factors related to both the quality of CAR-T cells administered and the immunosuppressive TME, which will be discussed in more detail below.

Fourth-generation CAR-T cells secreting transgenic IL-7, IL-12, IL-15, and IL-18 have been shown, in preclinical trials with solid tumors, to enhance T-cell activation and stimulate the recruitment of innate immunity cells to kill antigen-negative cancer cells in the lesion focus. For example, 4H11-28z/IL-12 CAR-T cells specific for the MUC-16ecto antigen have been generated. This antigen is overexpressed on most ovarian tumor cells and is a residue of MUC-16 after CA-125 cleavage. These CAR-T cells constitutively secreted IL-12 and demonstrated enhanced antitumor efficacy on Scid-Beige mice with human ovarian cancer xenografts, as determined by increased survival, prolonged T-cell persistence in the TME and higher systemic levels of IFN-γ [141,142,143]. In another case, to prevent systemic toxicity, CAR-T cells have been engineered with an expression cassette driven by the NFAT/IL-2 promoter to release inducible IL-12 upon CAR binding. Once the CAR interacts with the antigen, CAR-T cells produce IL-12, providing a constant high level of cytokines in the target organ; IL-12 production ceases when CAR-T cells cease contacting their cognate antigen, representing a safe “shutdown” after leaving the target tissue. Such a construct is known as TRUCKs. There are several other types of CAR-T cells. Universal CARs that have no endogenous TCR or MHC, self-destructive CARs (CXCR2-expressing CAR-T) carrying a chemokine receptor on their surface that binds to chemokines secreted by tumor cells, self-destructive CARs expressing inducible modified human caspase 9 fused to human FK506-binding protein which provides external signals to stop their activity), and methotrexate (MTX)-conditional CAR-T cells whose antitumor effect can be modulated by MTX are currently under investigation [131,142,144,145,146,147].

Clinical trials with EGFR-, mesothelin-, GD2-, and other second-, third-, and fourth-generation autologous CAR-T cells are underway (Table 1). Allogeneic CAR-T cells targeting MUC1C (P-MUC1C-ALLO1) are also undergoing Phase I clinical trials for the treatment of patients with advanced or metastatic solid tumors (NCT05239143).

**Limitations to CAR-T cell therapy in solid tumors and their solutions.** The low efficacy of systemic CAR-T therapy in solid tumors, as mentioned above, is attributed to several factors related to both immunosuppressive TME and the quality of administered CAR-T cells. Among them are difficulties in choosing appropriate TAAs, poor tumor infiltration by CAR-T cells, weak expansion and persistence in vivo, inherent T-cell dysfunction and depletion in immunosuppressive TME, intrinsic target antigen heterogeneity, or antigen loss by target cancer cells. In addition, short- and long-term side effects such as cytokine release syndrome (CRS), immune effector cell-associated neurotoxicity syndrome (ICANS), the potential for “on-target off-tumor” (OTOT) effects and graft-versus-host disease (GVHD in the case of donor CAR-T), and the high cost and lengthy process of producing modified T cells are also the challenges faced in the treatment of malignancies [47,120,148,149,150].

For T-cell therapy to be effective, the targeted tumor antigen must, as in the case of ADC, be expressed (a) on the surface and (b) only on cancer cells and not on healthy cells. Targeting TAAs that are also expressed in normal tissues results in severe CAR-T cell toxicity. The problem of TAA selection in solid tumors is solved by revealing and identifying neoantigens originating from tumor-specific gene mutations, since their formation and their expression are restricted to tumor cells. However, cell-surface neoantigens are rare, particularly in tumors with a low mutational burden [151].

There are several original ways to overcome CAR-T-cell-mediated OTOT toxicity. Logic-gated CAR-T cell approach is a reflection of the mathematical operators “IF/THEN”, “AND”, “OR”, and “NOT” that can be applied to increase the specificity of cell killing and reduce the potential for OTOT. For example, dual AND-logic CAR-T cells expressing two different recombinant receptors with different intracellular signalling domains were engineered to simultaneously target two different TAAs, CEA and mesothelin. These cells demonstrated specific killing of dual-TAA-positive cells without recognising single-TAA-positive cells in mice. This approach reduces the risk of OTOT because non-malignant tissues typically express only one of these TAAs, and the binding of only one receptor would result in the weakening of activating CAR-T cell signalling [152]. Another way to overcome CAR-T-cell-mediated OTOT toxicity has been proposed, which takes into account some features of TME (low oxygen levels, high levels of reactive oxygen species, and increased protease activity) in designing CARs. One example is the so-called masked CAR, which contains an N-terminal masking peptide that blocks the antibody-TAA binding site and a protease-sensitive linker. The TME has a high local concentration of proteases that can cleave the linker and release the masking peptide, thereby liberating the antigen-binding site of the CAR. This design allows CAR-T cells to remain inert when encountering antigens in healthy tissues and to be activated in the TME, allowing CAR-T cells to recognise target antigens only at the tumor site [153,154]. One other example of exploiting features of TME is hypoxia-sensitive CARs. In this case, CAR expression was absent at oxygen concentrations corresponding to healthy organs (≥5%) but appeared on the cell surface at oxygen concentrations equivalent to those present in TME (≤1%). These CAR-T cells demonstrated high antitumor activity in a mouse model of ovarian adenocarcinoma xenotransplantation without systemic toxicity [155]. Elevated levels of reactive oxygen species in TME are another feature of solid tumors. High levels of reactive oxygen species are well tolerated by tumor cells due to increased expression of antioxidant proteins, but this suppresses T-cell activity and viability. To empower the antioxidant capacity of CAR-T cells against pro-oxidant TME, CAR-T cells expressing either the enzyme catalase or thioredoxin-1 were generated. It was found that removal of reactive oxygen species in TME protected the injected T cells and allowed CAR-T cells to retain cytolytic immune synapse formation, cytokine release, proliferation and tumor cell killing properties under pro-oxidative conditions [156,157].

Heterogeneous expression of target antigens in solid tumors is another major reason for the failure of CAR-T cell therapy. Often after initial tumor shrinkage by CAR-T-cells, antigen-negative cancer cells not recognised by CAR provoke tumor relapse. During tumor progression and/or under the influence of chemotherapy, cancer cells may lose TAA by downregulating or mutating the target antigen, preventing their specific recognition even by tumor-infiltrating T cells. An important extratumoral effect of CAR-T-cell therapy is trogocytosis, which consists of receptor-mediated transfer of cognate antigen from tumor cells to immune cells expressing the receptor [158]. This results in loss of antigen and promotes tumor escape from CAR-T cell therapy [134]. Tumor antigen escape is the main reason of developing resistance to CAR-T cell therapy [134,158,159].

The choice of target TAA is a key factor determining the specificity and efficacy as well as survival of CAR-T cells. As mentioned above, in order to circumvent tumor resistance and enhance the effector action of CAR-T cells, CAR constructs that simultaneously target two or more different TAAs have been developed. This is achieved with such approaches as (1) combined CARs or multiple CARs—using two or more CAR-T cells to target different TAAs; (2) dual-signalling CARs—expression of two different CARs on the surface of a single T cell; (3) bispecific “tandem” CAR or TanCAR—two different scFvs join hand-in-hand to form a tandem CAR on the surface of a single T cell; and (4) separation of the costimulatory domains (e.g., CD28 and 41BB) from CD3ζ on two different CARs (referred to as split-signal CARs) [120,140,152,160]. Through these approaches, CAR-T cells are able to enhance their anti-tumor efficacy. This indicates that, unlike TriTEs and TriKEs, increasing the specificity of CAR-T cells leads to an expanded population of cells in the tumor that can be treated.

To improve T cell recruitment into the tumor, CAR-T cells were also engineered to target antigens such as integrins, which are selectively enriched in the TME, or vascular endothelial growth factor (VEGF) receptor-2 (VEGFR2), a highly expressed antigen enriched in the tumor vasculature. Interaction with appropriate CAR induced the process of T-cell release from the vascular network into tumor tissue, followed by the efficient antigen-specific killing of tumor cells [120,161,162]. Another approach to recruiting T cells into tumor is to use constructs expressing a chemokine receptor, such as CCR2, CCR2b, CXCR3, CCR4, CCR7, CXCR2, or CXCR4, which can interact with the relevant tumor-derived chemokine. However, the chemokine landscape in tumors can be extremely heterogeneous, so the identification of specific targets for different cancers, and for different patients, is necessary. In the case of injury or autoimmune disease, additional toxicity may occur due to non-tumoral production of chemokines that may “divert” T cells away from the putative tumor target. Alternatively, intratumoral delivery may help circumvent this problem. As an example, satisfactory results of regional infusions: intraperitoneal and intrahepatic delivery (via percutaneous hepatic artery) of anti-carcinoembryonic antigen (CEA) CAR-T cells, in appropriate colorectal cancer metastases in phase I clinical trials have been reported [163,164]. In patients with metastatic breast cancer with accessible metastases to the skin or lymph nodes, the intratumoral administration of mRNA-transfected cMet-CAR-T cells caused extensive tumor necrosis at the injection site and marked macrophage infiltration in preliminary studies [165]. A Phase I clinical trial of cMet-CAR RNA T cells also includes patients with operable triple negative breast cancer (NCT01837602). However, the location specificity of many tumors, technical difficulties in reaching the tumor focus, and site of metastasis make local delivery difficult and not always applicable [166,167].

The accumulation of ECM in the TME and poor tumor vascularisation contribute to increased interstitial pressure and impaired perfusion, which in turn leads to hypoxia [168,169]. Physical barriers stipulated by the tumor stroma, soluble products of impaired cellular metabolism in the tumor, an acidic environment and inhibitory proteins such as PD-L1 can inhibit the recruitment, invasion, activation, and persistence of CAR-T cells in tumor tissue, promoting the recruitment of immunosuppressor cells (Treg, MDSCs, and TAMs) and allowing tumor cells to avoid immune recognition. Then, TME is further enriched with immunosuppressive cytokines/growth factors, one of the main ones being TGF-β. High concentrations of TGF-β are found in a variety of solid tumors, accompanied by poor prognosis. TGF-β disrupts innate and adaptive cellular immunity, creating a favourable microenvironment for tumor growth and metastasis, and is considered an important component of tumor anti-immune defence. Elevated levels of TGF-β block the differentiation of naive T cells into effector T cells, promote their transition to the Treg subpopulation, and attenuate the antigen-presenting functions of DCs, contributing to further immune repression. TGF-β-mediated suppression is not limited to T cells, as this cytokine also suppresses many aspects of NK cell function, including cytokine secretion, degranulation and metabolism. T cells with CD28-ζ CAR, but not with 4-1BB-ζ CAR, have been shown to resist TGF-β-mediated repression. This is determined by Lck (lymphocyte-specific protein tyrosine kinase)-binding motif in CD28 CARs, whose activation leads to IL-2 release and resistance to TGF-β-mediated IL-2 receptor signalling. To confirm this, IL-2 deficient CD28ΔLCK-ζ CAR-T cells were engineered with a hybrid IL-7 receptor that provides IL-2Rβ chain signalling upon IL-7 binding. Such modified T cells demonstrated improved CAR-T cell activity against TGF-β+ tumors [170,171,172].

CAR-T cells targeting untransformed stromal cells, in particular CAFs, which promote the production of ECM components and growth factors (including TGF-β) and tumor progression, can suppress tumor growth and enhance host immunity without severe toxicity. FAP, a member of the serine protease family, is known to be expressed on the surface of CAFs. Adaptive FAP-CAR-T cell transfer, as shown by Lo et al., lowered the amount of ECM proteins and glycosaminoglycans, reduced tumor vessel density, suppressed the growth of desmoplastic human lung cancer xenografts and syngeneic mouse pancreatic cancers in an immune-independent manner, and inhibited the growth of autochthonous pancreatic cancer [173,174]. Fourth-generation CAR-Ts targeting Nectin-4 and FAP in the tumor stroma and releasing IL-7 and CCL19 or IL-12 upon CAR-target interaction are known in Phase I clinical trials for the treatment of solid tumors such as non-small cell lung cancer, breast cancer, ovarian cancer, bladder cancer, and pancreatic cancer (NCT03932565) (Table 1).

An approach to combining adoptive T-cell transfer with immune checkpoint inhibition is to genetically engineer CAR-T cells to inherently disrupt or downregulate immune checkpoint signalling. Moreover, by stably preventing immune checkpoint signalling, this approach could potentially evade the adverse effects associated with long-term use of pharmacological ICIs. Downregulation of PD-1, CTLA-4, and TIGIT using CRISPR/Cas9 or small interfering RNAs resulted in increased cytotoxicity of CAR-T cells, promoted a sustained antitumor response, and improved CAR-T cell persistence in vivo [120,175,176]. An ideologically similar but technologically distinct approach is to generate T cells secreting or carrying immune checkpoint-blocking antibodies. One recent example is the study of CAR-T cells with hyaluronidase and the checkpoint-blocking antibody α-PD-L1 on the surface of CAR-T cells. The modified hyaluronidase degraded hyaluronic acid and disrupted tumor ECM, allowing CAR-T cells to deeply infiltrate solid tumors. These cells demonstrated strong antitumor activity and significant therapeutic efficacy in two solid tumor models without significant systemic side effects [177,178]. Currently, CAR-T cells secreting antibodies against PD-1/CTLA-4/TIGIT are being evaluated in clinical trials (NCT03198546, NCT03198052) (Table 1). However, the downregulation of inhibitory molecules can lead to uncontrolled T-cell proliferation or increased risk of autoimmunity, and the use of exogenous cytokines can lead to the development of serious adverse effects. Thus, in late January 2024, the FDA recommended including new “black box” warnings for CAR-T-cell therapy after reviewing clinical trial reports describing the occurrence of mature CAR-positive T-cell malignancies following CAR-T-cell immunotherapy [179,180].

In addition to scientific challenges, there are currently two major limitations of CAR-T therapies: patient access and high cost. Access is a major limitation even in large academic, comprehensive cancer centres in developed countries where modern commercial drugs are centrally produced. Access may be limited by the CAR-T production capacity of the respective GMP manufacturers. Patients often wait up to 3 weeks for a designated date when a company can receive the patient’s autologous apheresis product to begin CAR-T cell production. In addition, single batch production requires the collection of initial apheresis cell material for each patient, and cell composition can vary, challenging pharmaceutical companies to standardise the manufacturing process as much as possible.

Another major access limitation is the prolonged manufacturing and release time that frequently ranges from 2 to 4 weeks. In addition, these cost and time-consuming therapies are associated with a risk of manufacturing failure of up to 25%. Due to the centralised nature of the process, transport and cryopreservation are required, which adds more time to the clinically relevant “vein to vein” time (time from leukapheresis to CAR-T infusion). Candidates for CAR-T therapy typically have an aggressive disease and undergo intensive pretreatment, and may often have a clinical deterioration that may make treatment too late.

The commercial production of CAR-T products is associated with significant costs, running into hundreds of thousands of dollars/euros. These costs place a heavy burden on the health care system of any country.

Solutions to overcome these problems include developing a more rapid manufacturing process, using off-the-shelf allogeneic CAR-Ts, and decentralising the production of autologous CAR-Ts [145,181,182,183,184,185]. Thus, at present, there are still many obstacles limiting the application of CAR-T cell therapy in solid tumors.

##### T Cells Secreting BiTes

A new approach combining the advantages of ACT and bsAbs has been proposed. Since both constructs, CAR and BiTE, can bypass the traditional TCR recognition of MHC-presented antigens on tumor cells, they can directly recognise the antigen on cancer cells. Therefore, combining both platforms may be a viable strategy to enhance the efficacy of CAR-T cell therapy in solid tumors, as has been shown in vitro and in mouse models of solid tumors. For example, Choi et al. developed a bicistronic construct to drive the expression of a CAR specific for EGFRvIII, a glioblastoma-specific tumor antigen, and BiTE against EGFR. CART.BiTE cells secreted EGFR-specific BiTEs that redirected CAR-T cells and recruited non-transduced anti-EGFR wild-type T cells [121].

Another interesting project of CAR-T cells secreting BiTE is directed against two tumor antigens that are highly expressed on ovarian cancer cells: cell surface Muc16 and intracellular WT1. CAR-T cells specific to Muc16 were engineered to secrete BiTE with specificity for both CD3 and WT1, represented by HLA-A2 molecules. Secreted BiTE recruited and redirected CD3+ T cells to WT1+/HLA-A2+ target cells, thereby enhancing cytotoxicity through targeting against dual antigens [186]. Thus, dual cytotoxic regimens with different specificity targeting surface and intracellular TAAs represent a promising strategy to overcome resistance to CAR-T cell therapy in solid cancers.

#### 3.2.2. NK Cell-Based Therapy

The success and limitations of existing immunotherapies have prompted researchers to develop alternative approaches and have attracted considerable attention to other cytotoxic immune cells, primarily those such as NK cells. NK cell immunotherapy may provide a new approach to overcome the limitations of T cell immunotherapy by activating the innate immune system without the need for antigen processing and presentation. To date, about 200 clinical trials of NK cells (autologous NK, allogeneic NK, and CAR-NK) and their activators for the treatment of solid tumors have been registered, 40 of which are clinically active and continue to recruit patients; this indicates that this field is being studied quite intensively.

##### Autologous NK Cells

Autologous NK-cell infusions were the first major focus of adoptive NK-cell therapy due to the convenience of using the patient’s own blood as a cell source, the lack of need for immunosuppressive therapy, and the low risk of GVHD reaction, a potentially fatal post-transplant condition caused by the destruction of host tissue by donor immune cells. In one of the first studies, NK cells were shown to be able to persist in the blood of patients for several months after infusion of autologous adaptively propagated in vitro NK cells in combination with high-dose IL-2 therapy. However, no clinical response was observed in any patient with stage IV melanoma or renal cell cancer. Prior to NK cell infusion, patients received non-myeloablative lymphodepleting chemotherapy with high-dose cyclophosphamide and fludarabine, which increased endogenous IL-15 levels to promote the persistence of donor NK cells in vivo [187]. Non-myeloablative lymphodepleting chemotherapy is used to deplete mature host lymphocytes that compete with the injected NK cells by acting as “cytokine sinks”, primarily IL-15. The lack of clinical benefit, possibly due to the inhibitory effect of the interaction between autologous NK cells and their own HLA molecules, has led many research groups to shift their attention from autologous NK cell therapy to allogeneic NK cell therapy [187,188].

##### Allogeneic NK Cells/Approaches to Adoptive NK Cell Therapy

Rapid NK cell therapy was initially achieved by isolating NK cells at the point of care through donor apheresis, activation, and fresh infusion. Newer methods allow differentiation and/or propagation of NK cells ex vivo, which are then cryopreserved for storage and administered to the patient after thawing at the patient’s bedside. NK cells are capricious cells in terms of transfection/transduction efficiency. Current methods of NK-cell modification are mainly based on mRNA transfection and transduction with viral vectors. However, the expression of exogenous mRNAs in NK cells is unstable, and transduction with viral vectors ends up with low efficiency in most cases. In fact, this is not unexpected since it is well known that NK cells are among the first responders to viral infections and must have undergone evolutionary selection to have high resistance to viral infection/foreign nucleic acids. To address this problem, new efficient and standardised protocols have been developed. Different transduction enhancers such as retronectin and vectofusin-1 are synthesised and selected. Alternatively, DNA electroporation offers a versatile platform to induce gene expression in NK cells. Recently, the development of genetic editing techniques such as the CRISPR/Cas9 system has opened new perspectives for precise genetic customisation [184,189,190,191,192].

There are several approaches to obtaining allogeneic NK cell therapies for rapid point-of-care or off-the-shelf use, which include (a) purified NK cell therapies, (b) expanded NK cell therapies, and (c) CAR-NK cell therapies. The off-the-shelf allogeneic NK cell therapies are available very rapidly, eliminating the need for multi-week production, as in the case of autologous CAR-T cells, and significantly cutting costs by allowing multiple patients to be treated with NK-cell therapies from a single donor. In addition, NK cells possess dozens of activating and inhibitory receptors, greatly minimising the likelihood of failed antigen recognition.

##### NK Cell Sources

Potential sources of NK cells include peripheral blood mononuclear cells (PBMCs), haematopoietic stem and progenitor cells (HSPCs) collected from umbilical cord blood (UCB), peripheral blood and bone marrow (BM), NK cell lines, human embryonic stem cells (hESCs), and induced pluripotent stem cells (iPSCs). Obtaining NK cells from BM is onerous and associated with potential risks to healthy donors [193,194].

Peripheral Blood and Umbilical Cord Blood NK Cells

To date, most adoptive NK cell therapies have used readily available sources: peripheral blood NK cells and umbilical cord blood NK cells. However, the number of NK cells contained in collected cord blood or peripheral blood is insufficient to achieve clinical therapeutic effects. To efficiently propagate NK cells ex vivo, Yonemitsu et al. established a technique for culturing highly activated NK cells with purity ≥ 90% from PBMCs in a completely closed system without nutrient devices. Another example is a method to increase CD34+ HSCs from cryopreserved cord blood 2000-fold and obtain a product consisting of 90% functional NK cells. This process takes place in closed large-scale bioreactors, which makes it a valuable tool for obtaining clinical-grade NK cells for adoptive transfer [195,196].

It is worth mentioning that each of these cell sources has significant drawbacks, including problems related to cost, delays in blood harvesting, donor variability, and heterogeneity of leukocytes in donor blood. In addition, despite great advances in techniques for isolation, purification, transduction, and proliferation of primary NK cells, difficulties remain. NK cells are often heterogeneous populations that have different stages of maturation and different expression patterns of activating and inhibitory receptors [91,188]. Along with this, preclinical and clinical studies have demonstrated the safety and efficacy of these NK cells against a variety of haematological malignancies and solid tumors. A number of Phase I clinical trials of allogeneic blood-derived NK cells (SNK01 and SNK02) as monotherapy for the treatment of solid tumors refractory to conventional therapy are known to be ongoing (NCT03941262, NCT05990920).

NK cell lines

Cell lines derived from NK cells are YT, YTS, NK-92, NKL, NK-YS, KHYG-1, IMC-1, NKG, and other cells, which can readily and unrestrictedly proliferate in vitro [197,198,199,200,201,202,203,204]. Immortal NK cell lines with inherent chromosomal abnormalities and a risk of malignant transformation require irradiation before administration to patients. It suppresses their proliferation and persistence in vivo while preserving their cytotoxic activity [91].

Among NK cell lines, the most widely used in clinical practice is the immortalised IL-2 dependent line NK-92, derived from peripheral blood mononuclear cells of a patient with rapidly progressive non-Hodgkin’s lymphoma. NK-92 cells were shown to express almost all major NK-activating receptors, lacking only surface expression of CD16. In addition, they express only the killer-cell immunoglobulin-like receptor (KIR) of the KIR2DL4 family and none of the other KIR family members that are characterised as inhibitory receptors. Activating receptors such as NKp30 were found to be strongly expressed on the surface, while NKp44 is expressed at a lower level. Given their broad receptor repertoire, as well as their IFN-γ secretion and cytotoxic abilities, NK-92 are considered the most “NK cell-like” cell lines in clinical and research practice. This also makes them a desirable model for research due to their ability to be transformed and cultured much more efficiently than primary NK cells. However, although adaptive transfer of irradiated NK-92 cells has demonstrated safety and preliminary evidence of clinical benefit in cancer patients, irradiation limits the in vivo survival of these cells to a maximum of 48 h, a potential barrier to long-term clinical efficacy [188,205,206,207].

The NK-92-derived line haNK was engineered to express CD16a and endogenous IL-2 restoring ADCC capability and obviate the need for culturing with exogenous IL-2, respectively. In addition, IL-2 can replenish NK cell granule stores, leading to the enhanced perforin- and granzyme-mediated killing activity of “depleted” NK cells [208]. HaNK cells also require lethal irradiation prior to any clinical use. The haNK cell line in combination with avelumab (anti-PD-L1 mAbs) and the cancer vaccine NANT and/or IL-15 is currently undergoing clinical trials in patients with solid malignancies such as relapsed/refractory advanced stage triple negative breast cancer (NCT03387085), squamous cell carcinoma (lung or head and neck) that has progressed after platinum-based chemotherapy and ICI (NCT03387111), and relapsed/refractory pancreatic cancer (NCT03586869).

Induced pluripotent stem cell-derived NK cells

A promising therapeutic alternative to NK cells from blood and NK cell lines are iPSC-derived NK cells, which can be used as a standardised “off-the-shelf” treatment for any patient, regardless of HLA haplotype. This not only allows for virtually unlimited numbers of homogeneous NK cells, but also makes these cells much more amenable to genetic manipulation than primary NK cells or NK cell lines. Experiments in mice bearing ovarian cancer xenografts have shown that treatment with iPSC-derived NK cells and blood-derived NK cells leads to a similar increase in the average survival of the mice. Due to their ability to readily differentiate into NK cells and their long-term expansion potential, iPSCs can be used to generate a great quantity of well-defined NK cells that can be stored “on-the-shelf” and used for therapy for large numbers of patients, including multiple dose treatment as needed [209,210,211].

Recently, FT500, a first-in-class off-the-shelf iPSC-derived NK-cell therapeutic in combination with ICI and IL-2, has undergone a Phase I clinical trial in patients with advanced solid tumors (NCT03841110).

##### CAR-NK Cells

The impressive clinical results of CAR-T cells, on the one hand, and the serious hurdles to CAR-T immunotherapy, especially in solid tumors, on the other hand, have encouraged research using CAR-NK cells. CAR-NK cells have several advantages over CAR-T cells. First, unlike CAR-T cells, CAR-NK cells retain their inherent ability to recognise and target tumor cells through their native receptors, preventing tumor cell escape when the target antigen CAR is suppressed. Second, CAR-NK cells do not undergo clonal expansion or immune rejection within weeks, and therefore do not cause safety concerns such as the CRS intrinsic to CAR-T-cell treatment. Finally, NK cells do not require strict HLA matching and, as noted above, do not induce GVHD, an important risk associated with CAR-T-cell immunotherapy, allowing CAR-NK cells to become a ready allogeneic therapeutic agent.

CARs for NK cells are mainly based on the first three generations of CARs. Their structure, like that of CAR-T cells, includes three components: an extracellular antigen-binding domain (usually scFv derived from mAb) and spacer, a transmembrane domain, and an intracellular activation domain. To enhance the antitumor efficacy of CAR-NK cells, several studies have proposed to enrich CAR with domains such as NK-specific 2B4 (CD244), DAP10, or DAP12 as signalling adaptor molecules involved in the signal transduction of activating NK cell receptors. For example, Li et al. developed a CAR targeting mesothelin, a cell surface antigen overexpressed in many solid tumors, and containing an NKG2D transmembrane domain, a 2B4 co-stimulatory domain and a CD3ζ signalling domain to mediate strong antigen-specific NK cell signalling. In a mouse model of human ovarian cancer xenograft, iPSC-derived NK cells expressing this CAR inhibited tumor growth and prolonged survival compared to peripheral blood NK cells, iPSC-derived NK cells, or CAR-T cells expressing the same construct. This difference is likely to make it feasible to treat patients with multiple doses of CAR-NK cells, which may lead to better clinical outcomes compared to the single dose that is commonly used for CAR-T-cell therapy in view of its limited cell availability and high cost [212]. In another study, CAR containing prostate stem cell antigen (PSCA) scFv fused to DAP12 CAR provided enhanced cytotoxicity of the NK cell line YTS against PSCA-positive tumor cell xenografts and resulted in complete tumor eradication in a significant proportion of treated mice [213]. Other cell sources, including primary NK cells and other NK cell lines, have also been probed to generate CAR-NK cells and evaluated in in vitro and in vivo models (Table 2).

Each of the sources described above has unique advantages and disadvantages for producing scalable and clinically relevant doses of NK cells. Some authors believe that primary NK cells are not an ideal substrate for CAR-cell products due to the challenges encountered in cell isolation, transduction, and propagation. This is partly supported by the fact that most current clinical trials of CAR-NK cells focus on NK-92 cell-derived and iPSC-derived products [134].

Thus, a preclinical evaluation of iPSC-derived NK cells expressing CAR targeting the conserved α3 MICA/B domain and a shedding-resistant form of the CD16 receptor was performed. The MICA and MICB family stress proteins are widely expressed by tumor cells after DNA damage, but are rapidly lost to avoid immune detection. These cells demonstrated potent antigen-specific cytolytic activity in vivo against solid and haematological xenograft models, which was further enhanced when combined with tumor-targeted therapeutic antibodies to provide ADCC [220]. However, a Phase I clinical trial of FT536, a first-in-class iPSC-derived MICA/B-targeted CAR NK cell therapy for advanced solid tumors, has been discontinued without disclosure (NCT05395052).

There are currently more than 25 clinical trials of CAR-NK cells against solid tumors [220,221,222]. Table 3 presents only selected examples of these studies. For example, Li et al. presented a case report of Robo1-specific CAR-NK cell therapy for the treatment of PDAC with liver metastasis. Robo1 is a member of the immunoglobulin superfamily and is associated with angiogenesis of various tissues, particularly pancreatic cancer. A CAR construct containing an extracellular domain against Robo1, a transmembrane domain (CD8), and the CD3 ζ and 4-1BB costimulatory molecules was expressed in NK-92 cells. The patient received an infusion of Robo1-specific CAR-NK cells, and liver metastases were treated by percutaneous injection of the same cells. No significant side effects were observed. The overall patient survival was 8 months [223]. Another example of CAR-NK-92 therapy is PD-L1 t-haNK cells. PD-L1 t-haNK cells are an off-the-shelf derivative of NK-92 cells engineered to express CARs targeting PD-L1. PD-L1 t-haNK cells are currently undergoing Phase I/II clinical trials in patients with advanced or metastatic solid cancer (NCT04050709, NCT04390399).

**Challenges to NK cell therapy in solid tumors and their mitigation strategies**. Obstacles to the use of NK cell therapy are related to: (1) the sensitivity of NK cells to the freezing and thawing process, as, depending on the regimes used, NK cells, to a greater extent than T cells, lose activity after thawing; (2) ex vivo expansion of NK cells to clinical levels (except in NK cell lines); (3) the possibility of contamination of NK cell preparations with T or B cells, which could theoretically cause GVHD or post-transplant lymphoproliferative disease, respectively; (4) limited persistence of NK cells in vivo; and (5) immunosuppressive effects of TME in solid tumors (e.g., limited infiltration by NK cells, tumor evasion by NK cell activity). There are various strategies to overcome these problems, such as ex vivo preconditioning with cytokines and/or small molecular drugs, creating “off-the-shelf” or iPSC-derived CAR-NK cells [224,225,226].

Cytokines such as IL-2, IL-7, IL-15, and IL-21 enhance NK cell survival ex vivo or in vivo. However, IL-2 infusion has significant side effects, including fever, chills, myalgia, and capillary leak syndrome, and can also increase Treg levels, which suppresses NK cells. IL-15, although not contributing to Treg increase, can lead to dose-dependent toxicity, including neutropenia, when administered exogenously by bolus injection. Alternative approaches to exogenous cytokine administration include treating patients with lymphodepleting chemotherapy prior to NK cell infusion or inserting IL-2 or IL-15 genes into the CAR construct for ongoing cytokine support of CAR-transduced cells [225,227,228,229,230].

To improve CAR-NK-cell trafficking to the tumor site, CAR constructs include chemokine receptors specific for chemokines secreted by the tumor. For example, the chemokine receptor CXCR4 in EGFRvIII-CAR-NK cells promoted specific chemotaxis to glioblastoma cells secreting C-X-C chemokine ligand (CXCL)12/SDF-1α [231]. In another study, NKG2D-specific CAR-NK cells modified with CXCR1 demonstrated enhanced in vitro migration into tumor supernatants and increased tumor infiltration in vivo in human subcutaneous and intraperitoneal ovarian cancer xenograft models [232]. Incorporation of another chemokine receptor, such as CXCR3, into the CAR construct can induce NK cell migration towards CXCL9, CXCL10, and CXCL11 gradients [233].

In contrast to successful adaptive NK-immunotherapy of haematological malignancies, NK-immunotherapy of solid tumors encounters peculiarities of their TME. The TME of solid tumors has immunosuppressive activity that reduces tumor tissue infiltration by NK cells, inhibits the survival and persistence of NK cells in tumor tissue, strongly weakens NK-mediated cytotoxicity, and provides tumor escape from immune surveillance. The combined action of TME and cancer cells in solid tumor leads to NK-cell dysfunction by (1) decreasing activating signals and (2) increasing inhibitory signals for NK cells. Downregulation of activating receptors is associated with reduced expression of respective ligands on the surface of cancer cells, cancer cells production of soluble ligands blocking recognition by NK cell activating receptors, and inhibition of expression/function of NK cell receptors such as NKG2D, NKp46, NKp30, DNAX accessory molecule-1 (DNAM-1/CD226), and CD16 (FCγRIII). Enhanced inhibitory signalling is associated with NK cells receiving inhibitory signals from cancer cells that retain HLA expression and/or overexpression of inhibitory receptors such as PD-1, PD-L1, NKG2A, and T-cell immunoglobulin and ITIM domain (TIGIT). Tumor-mediated suppressive factors such as extracellular TME stimuli (hyaluronan accumulation, vascular collapse, hypoxia, low pH, adenosine-rich medium, cytokines, TGF-β, prostaglandin-E2, IL-dependent STAT3 activation, etc.) and suppressive immune cells (Treg, MDSCs and TAMs) also contribute to the inhibition of NK cell activity [206,226,227].

Treg, MDSCs, and TAMs are the dominant immune cell types that suppress NK cell activation. For example, a latent form of TGF-β consisting of mature TGF-β noncovalently associated with its amino-terminal propeptide can be cleaved by TAM protease to form the active form of TGF-β. TGF-β, as mentioned above, is a highly enriched growth factor in the TME of solid tumors. TGF-β suppresses the expression of NKG2D and NKp30, reduces the production of IFN-γ, and prevents the cytotoxicity of NK cells, inhibiting their antitumor activity [226,227,234,235,236]. TGF-β suppresses NK cell responses in TME and promotes the differentiation of NK cells into innate lymphoid cell-1-like type with abrogated effector function [237]. TGF-β also significantly enhances the expression of chemokine receptors, CXCR3 and CXCR4, characteristic of immature CD56brightCD16- NK cells, while inhibits CX3CR1 in CD56dim NK cells, which reduces their cytotoxicity [206,238,239]. In contrast, the neutralisation of TGF-β1 by mAbs can fully restore NKG2D expression as well as the cytotoxic functions of NK cells [240]. MDSCs can inhibit ADCC of NK cells through nitric oxide secretion, and Tregs can directly suppress NK cell function or indirectly inhibit NK cell activity by secreting IL-10 [206]. The remodelling of TME by enzymatic degradation of hyaluronan promoted the uptake of therapeutic antibodies into TME, increased the accumulation of T cells and NK cells, and decreased the number of MDSCs [241].

Thus, the advantages of NK cells over T cells include the greater availability and efficacy of NK cell therapy, including the ability to use allogeneic cells, a much lower risk of side effects, a faster response, and a stronger effect on other immune cells via secreted cytokines to enhance the anti-tumor response. However, many questions remain open, including figuring out the optimal source of NK cells, the dose, the administration schedule, cytokine support, and a better understanding of the mechanisms of response and resistance [184].

#### 3.2.3. CAR-Macrophages as an Innovative Platform for Anticancer Therapy

Unlike lymphocytes, macrophages can actively infiltrate solid tumor tissues and interact with virtually all cellular components of TME [242]. However, although macrophages are the most abundant cell type in the TME, due to an ability of tumor cells to be “domesticated”, they are unable to recognise and attack tumors. Even within tumor tissues, macrophages are not active in phagocytosis or antigen presentation, but instead switch to the immunosuppressive M2 type, which impedes the anti-cancer response of the immune system [243,244]. Studies pointing out the features of macrophages in TME have sparked interest in developing therapies aimed at depleting immunosuppressive TAMs, redirecting TAM polarisation towards the M1 phenotype through the secretion of pro-inflammatory factors and chemokines, and enhancing the phagocytic activity of TAMs. These approaches have shown promising potential for clinical application; active clinical trials of TAMs for the treatment of solid tumors are summarised in a review [46].

CAR-macrophage (CAR-M) cell therapy is now being considered as a new therapeutic option to address the problems associated with CAR-T cell therapy [244]. The structure of CARs for macrophages and T cells is quite similar, but it differs in the intracellular domain. A new class of synthetic receptors CARs for phagocytosis (CAR-Ps) was developed. Among them, constructs with cytosolic domains of multiple epidermal growth factor like domains 10 (Megf10) and FcRγ were selected as the most active triggers of engulfment. Megf10 indirectly recognises phosphatidylserine on the surface of apoptotic cells, while FcRγ is a specific classical macrophage signaling molecule involved in antibody-dependent cellular phagocytosis (ADCP); both have cytosolic ITAMs whose phosphorylation stimulates phagocytosis. Similar results were obtained with a CAR construct carrying the CD3ζ subunit of TCR containing three ITAM motifs [245]. In another study with human ovarian cancer cells, CAR-M cells also containing an intracellular CD3ζ signalling domain were tested. The created anti-HER2 CAR-M cells expressed pro-inflammatory cytokines and chemokines, converted bystander M2-macrophages to M1, enhanced antigen presentation, recruited T cells, and were resistant to the action of immunosuppressive cytokines. In two mouse models of solid tumor xenografts, CAR-M infusion reduced tumor size and prolonged overall survival. In humanised mouse models, CAR-M has also been shown to induce inflammation in TME and increase the activity of anti-tumor T cells [246,247]. Similar results were obtained with CAR-M targeting CD47 and carrying an intracellular CD3ζ domain. The engineered CAR-Ms exhibited antigen-specific phagocytosis of ovarian cancer cells in vitro and could activate CD8+ CTLs to secrete various antitumor factors. In an in vivo model, CAR-Ms potentiated the activation of CD8+ T cells, rerouted the phenotype of TAMs, and led to tumor regression [248].

CAR-M cells are typically obtained in two ways: ex vivo and in vivo from various cell sources (PBMCs, iPSCs, cell lines, HSPCs collected from bone marrow, cord blood, and peripheral blood) [248,249]. The ex vivo approach requires the selection, activation, expansion, and differentiation of CAR-M cells. In the in vivo approach, the CAR transgene mixes with nanoparticles to form nanocomplexes, which are then injected into the organism to convert TAM to CAR-M cells in the TME. Macrophages are very resistant to genetic engineering due to their high ability to detect and destroy foreign nucleic acids, but the development of genetic manipulation allows bypassing this by both viral and non-viral methods [250,251,252].

PBMCs are a readily available source of monocytic cells. Cells derived from such a source retain their primary morphology and can produce multiple pro-inflammatory factors such as IL-6, IL-8, and tumor necrosis factor alpha (TNF-α), as well as higher levels of surface markers including natriuretic peptide receptor (NPR), CD14, and CD68. However, limitations to the widespread use of primary macrophages in the clinic are the complicated genetic manipulation and the low yield of macrophages from PBMCs [247]. PBMCs can be reprogrammed into iPSCs and then genetically modified into CAR-iPSCs. iPSC-derived CAR-macrophage cells have demonstrated antitumor effects in vitro and in vivo in solid cancer models. In addition, Zhang and colleagues demonstrated the generation of iPSCs with anti-GD2 CARs and their differentiation into CAR-M cells through arterialised hemogenic endothelium intermediates. The resulting anti-GD2 CAR macrophages thus obtained exhibited strong cytotoxic activity against GD2-expressing neuroblastoma and melanoma in vitro and neuroblastoma in vivo [253,254].

The human cell lines U-937, THP-1, and Mono Mac 6, which are relatively immature monocytic-macrophage cells, are also considered as a source of therapeutic macrophages [255]. The THP-1 cell line was used to differentiate into macrophages and then, as well as the Mono Mac 6 cell line, was modified with anti-CEA CARs. These CAR-Ms exhibited antigen-directed phagocytic activity against target cells in vitro [249].

It was shown that CAR-M cells can recognise tumor cells through interaction with TAAs and respond by secreting various pro-inflammatory cytokines, inducing phagocytosis and promoting the remodelling of tumor ECM by matrix metalloproteinases. Activated CAR-M cells can stimulate other immune cell types such as TAMs, DCs, T cells, and NK cells and enhance their anti-tumor function [250].

Patient recruitment is currently underway for CT-0508 (anti-HER2 CAR-Ms), a multicentre Phase I clinical trial targeting patients with recurrent or metastatic solid tumors with HER2 overexpression whose cancer is not approved for HER2-targeted therapy or is not responding to treatment (NCT04660929).

The features, advantages, and disadvantages of CAR-T, CAR-NK, and CAR-M cells are summarised in Table 4. Potential strategies for optimising CAR-T, CAR-NK, and CAR-M therapy are discussed in more detail in a recent review [222].

It should be noted that other immune cells such as CAR-NKT and CAR-neutrophils may also be useful for immunotherapy of solid tumors. According to preliminary data, their use has yielded tangible results in in vitro and in vivo studies [256,257].

Different approaches to targeted immunotherapy in solid tumors are summarized in Figure 7.

## 4. Conclusions

Tumor immunotherapy has greatly improved the chances of successfully fighting cancer. The first achievements gave a huge boost to research in this direction. However, further research has shown that there are a number of issues that significantly reduce the effectiveness of immunotherapy, especially in solid tumors. First of all, these problems are related to the protective mechanisms of the tumor and its microenvironment. Currently, major efforts are focused on overcoming the protective mechanisms by using different ACT variants and modifications of genetically engineered constructs. Heterogeneity of both tumor structure and its microenvironment is often a decisive factor in the low efficacy of the applied immunotherapeutic agents. In patients with more monomorphic forms of solid tumors, it is expected that one or another immunotherapy option will be more successful. Therapy of heterogeneous solid tumors with prevailing TME likely requires a combination of therapeutic agents with a tailored approach depending on the morphological pattern of the particular tumor.

Finally, we took the liberty to formulate a simple “3C rule” that hopefully will be beneficial for immunotherapeutic approaches to solid tumors:

1. Consolidate histological subtyping with molecular genetic analysis to predict the patient response to treatment. This allows the targeting of therapies to specific mechanisms of carcinogenesis and, accordingly, the prescribing of effective therapies suitable for specific patients.

2. Crack the defence systems of both cancer cells and the TME by simultaneous treatment, targeting the cancer cells themselves, and cancer-associated fibroblasts and extracellular matrix proteins in the TME. This will contribute to better overcoming the protective mechanisms of both cancer cells and the TME. It is recommended that cellular immunotherapy be used concurrently with small molecule treatment to eliminate cancer cells before they develop resistance to the cytotoxic action of small molecules.

3. Cogitate on the interactive use of several types of CAR-immune cells (CAR-T, CAR-NK, CAR-M, CAR-NKT, etc.) in parallel or in series. This is expected to compensate for the limitations of each and to produce an immune response as close to the natural one as possible.

Thus, a variable immunotherapy may contribute to solving problems in the treatment of solid tumors.

## Figures and Tables

**Figure 1 cancers-16-02270-f001:**
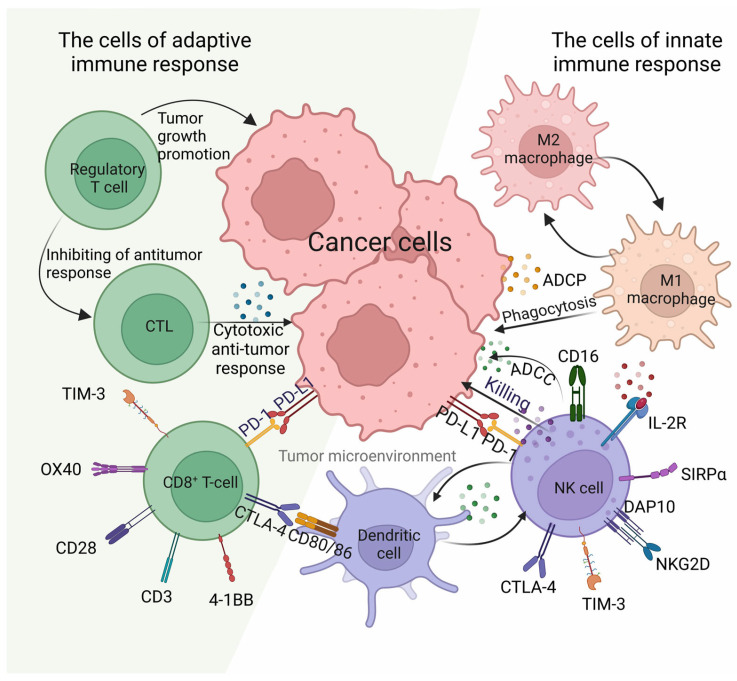
Schematic illustration of immune surveillance of the tumor microenvironment. 4-1BB, also known as CD137, a member of the tumor necrosis factor receptor superfamily T cell costimulatory receptor; ADCC, antibody-dependent cell-mediated cytotoxicity; ADCP, antibody-dependent cellular phagocytosis; CTL, cytotoxic lymphocyte; CTLA-4, cytotoxic T lymphocyte-associated protein 4; IL-2R, interleukin-2 receptor; NK, natural killer; NKG2D, natural killer group 2 member D protein (activating receptor); OX40, also known as CD134, tumor necrosis factor receptor superfamily, member 4 (TNFRSF4); PD-1, programmed cell death protein 1; PD-L1, programmed cell death ligand 1; SIRPα, signal regulatory protein alpha; TIM-3, T cell immunoglobulin and mucin domain-containing protein 3. This image was created with BioRender (https://biorender.com/).

**Figure 2 cancers-16-02270-f002:**
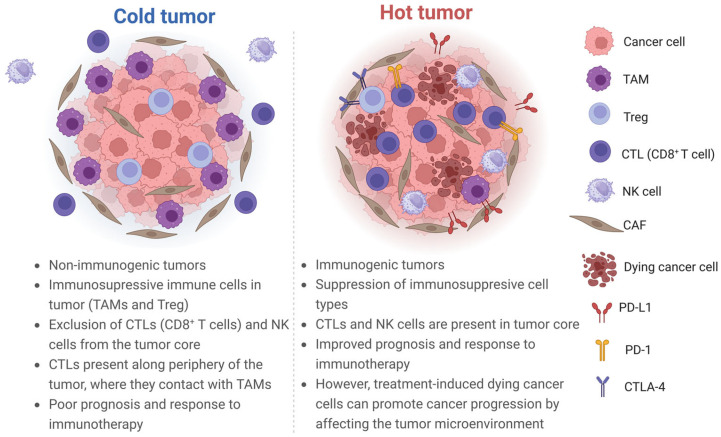
Difference between cold and hot tumors. In cold tumors, anti-inflammatory cells such as TAMs (macrophages of type 2, M2), one of the subpopulations of myeloid-derived suppressor cells (MDSCs), and regulatory T cells (Treg) are upregulated, but pro-inflammatory cells such as cytotoxic T lymphocytes (CTLs) are downregulated. In hot tumors, macrophages of type 1 (M1) and infiltrating lymphocytes (TILs) such as CTLs, effector T cells, and NK cells, are upregulated and tertiary lymphoid tissue can also be observed. Furthermore, hot tumors can be characterised by the presence of a high immunoscore of checkpoint activation, such as PD-1, PD-L1, CTLA-4, and others, a high tumor mutational burden, specific gene mutations (e.g., KRAS, TP53), and microsatellite status. CAF, cancer-associated fibroblast; CTLA-4, cytotoxic T lymphocyte-associated protein 4; PD-1, programmed cell death protein 1; PD-L1, programmed cell death ligand 1. This image was created with BioRender (https://biorender.com/).

**Figure 3 cancers-16-02270-f003:**
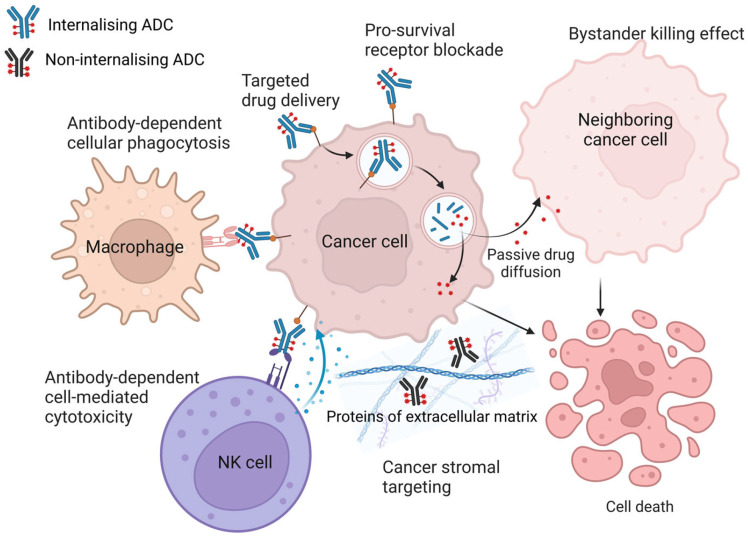
Targeting and mechanisms of ADC action. A detailed description is given in the text. This image was created with BioRender (https://biorender.com/).

**Figure 4 cancers-16-02270-f004:**
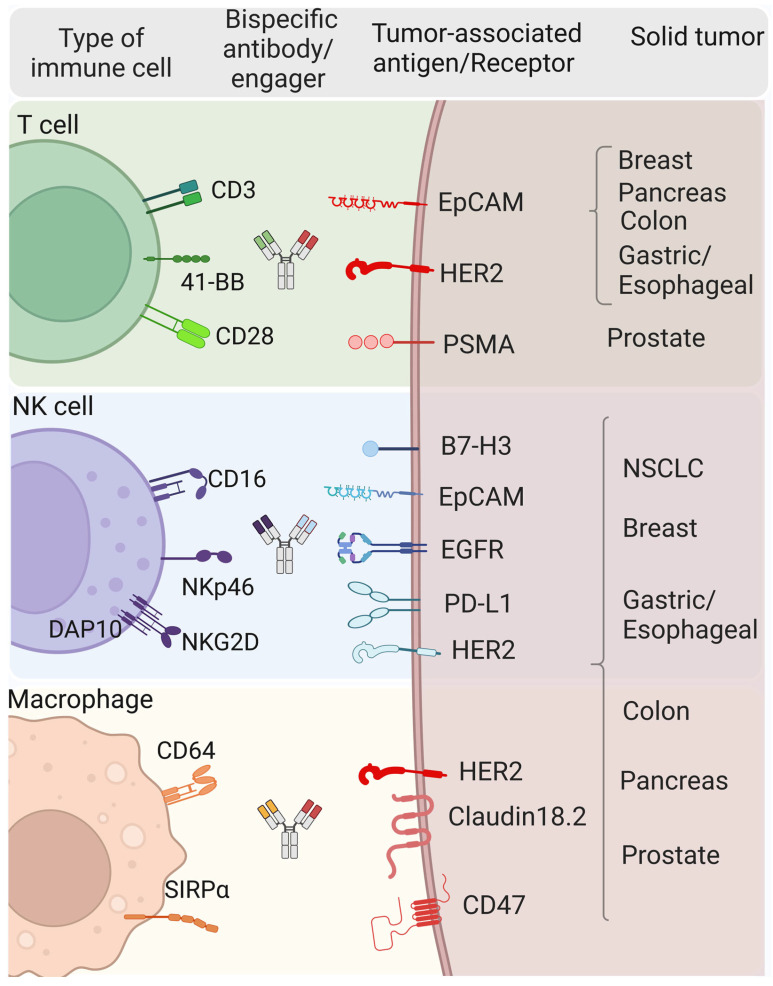
Schematic illustration of the bispecific immune cell engagers that redirect immune cells such as T cells, natural killer (NK) cells, and cytotoxic/phagocytic cells against solid tumor cells. See the text for further details. NSCLS, non-small cell lung cancer; PSMA, prostate-specific membrane antigen. This image was created with BioRender (https://biorender.com/).

**Figure 5 cancers-16-02270-f005:**
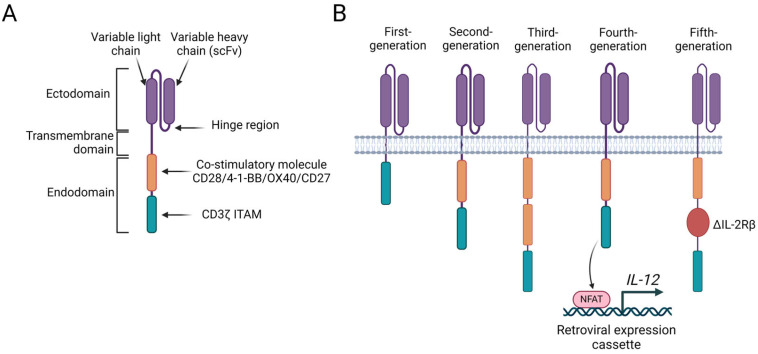
Structural details of chimeric antigen receptor (CAR)-T therapy: (**A**) Structure of a CAR; (**B**) Structural design of different generations of CARs. A detailed description is given in the text. ITAM, immunoreceptor tyrosine-based activation motif. This image was created with BioRender (https://biorender.com/).

**Figure 6 cancers-16-02270-f006:**
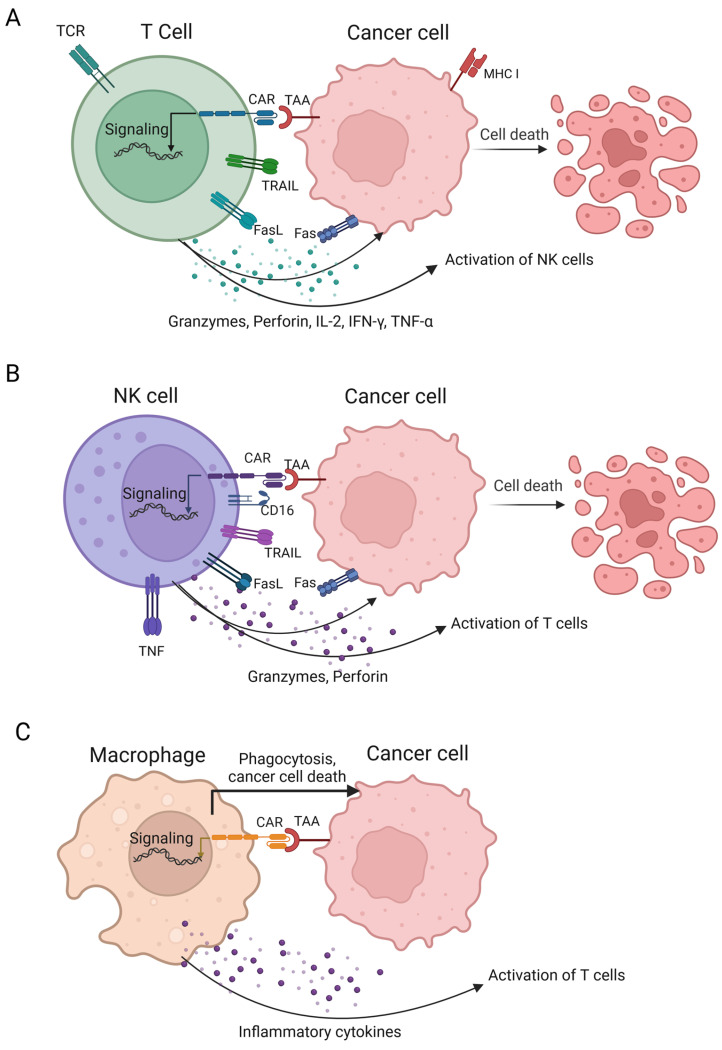
Schematic illustration of killing mechanisms of CAR-T (**A**), CAR-NK (**B**) and CAR-M (**C**) cells. This image was created with BioRender (https://biorender.com/).

**Figure 7 cancers-16-02270-f007:**
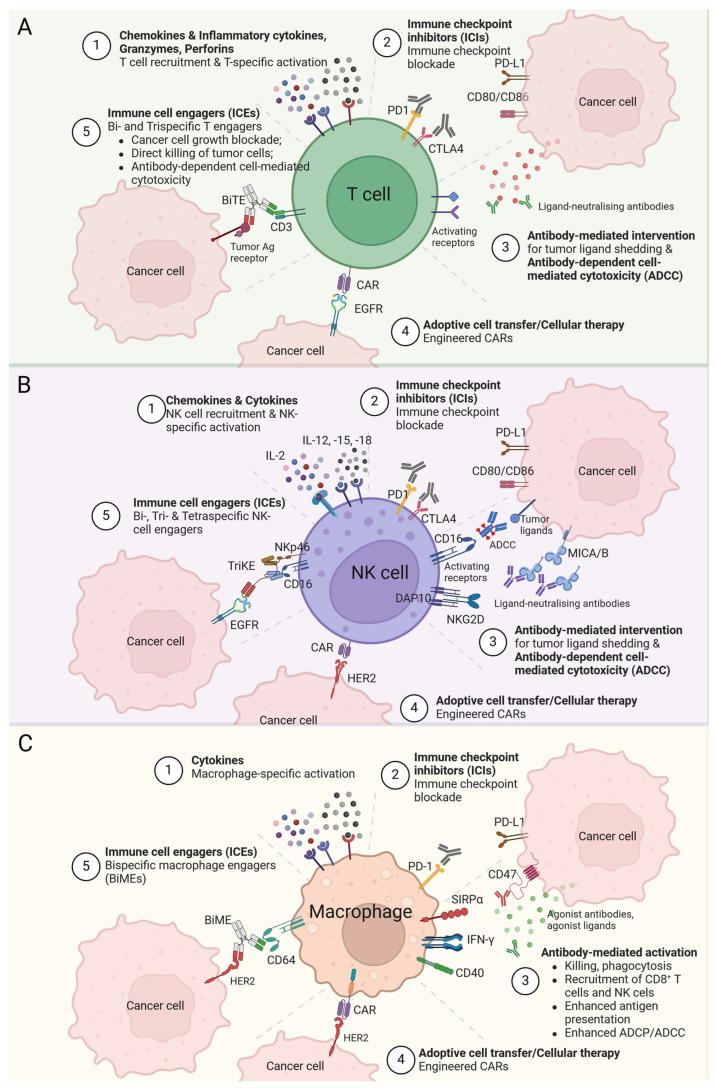
Different strategies for immunotherapy of solid tumors: (**A**) T cell-based approaches; (**B**) NK cell-based approaches; (**C**) Macrophage-based approaches. This image was created with BioRender (https://biorender.com/).

**Table 1 cancers-16-02270-t001:** Clinical trials of CAR-T cell therapy in solid tumors.

CAR-T Product	Target Antigen	Generation of CAR-T Cell Structure	Cancer	Trial Phase	Status	NCT Number
HER2-E-CART cells	HER-2	2nd-generation	HER-2-positive and refractory advanced solid tumors	I	Not yet recruiting	NCT05745454
Mesothelin/GPC3/GUCY2C-CAR-T Cells	Mesothelin	2nd-generation (secreting a fusion protein of IL21 and scFv against PD1)	Pancreatic cancer	I	Recruiting	NCT05779917
CART-TnMUC1 cells; Cyclophosphamide; Fludarabine	TnMUC1	2nd-generation	Advanced TnMUC1-positive solid tumors (triple negative breast cancer, epithelial ovarian cancer, pancreatic cancer, and non-small cell lung cancer), and advanced TnMUC1-positive multiple myeloma	I	Terminated	NCT04025216
LeY-CAR-T	Lewis Y Antigen (LeY)	2nd-generation	LeY antigen-expressing advanced solid tumors	I	Completed	NCT03851146
GPC3/Mesothelin/Claudin18.2/GUCY2C/B7-H3/PSCA/PSMA/MUC1/TGFβ/HER2/Lewis-Y/AXL/EGFR-CAR-T Cells	GPC3, Mesothelin, Claudin18.2, GUCY2C, B7-H3, PSCA, PSMA, MUC1, TGFβ, HER2, Lewis-Y, AXL, or EGFR	3rd-generation	Lung cancer	I	Recruiting	NCT03198052
Anti-HLA-G CAR-T cells (IVS-3001); Fludarabine phosphate; Cyclophosphamide	Human leukocyte antigen (HLA-G)	3rd-generation	Previously treated, locally advanced, or metastatic solid tumors that are HLA-G positive	I/IIa	Recruiting	NCT05672459
EGFR-IL12-CART Cells	EGFR	4th-generation (IL12 expressing)	Metastatic colorectal cancer	I/II	Unknown status	NCT03542799
anti-CTLA-4/PD-1 expressing EGFR-CAR-T	EGFR	4th-generation (CTLA-4 and PD-1 antibodies expressing)	EGFR-positive advanced solid tumor	I/II	Unknown status	NCT03182816
4SCAR-GD2 T-cells	GD2	4th-generation (with an inducible caspase 9 suicide gene)	Solid tumor	I/II	Unknown status	NCT02992210
C7R-GD2.CART Cells	GD2	4th-generation (IL7 expressing)	Relapsed or refractory neuroblastoma and other GD2 positive cancers (sarcoma, uveal melanoma, phyllodes breast tumor, or another cancer)	I	Active, not recruiting	NCT03635632
CAR-T therapy	Nectin4/FAP	4th-generation (IL7 and CCL19, or IL12 expressing)	Nectin4-positive solid tumors such as non-small cell lung cancer, breast cancer, ovarian cancer, bladder cancer, or pancreatic cancer, and FAP-positive CAFs in the tumor-associated stroma	I	Unknown status	NCT03932565
GPC3/TGFβ-CART cells	GPC3/ soluble TGFβ	3rd/4th-generation	Hepatocellular carcinoma with GPC3 expression, squamous cell lung cancer	I	Unknown status	NCT03198546

**Table 2 cancers-16-02270-t002:** Examples of NK cell sources for CAR-NK cells (according to [91], modified).

NK Cell Type	Target Antigen	Signalling Domain	Cancer	Gene Transfer Method	References
PBMC-NK	HER-2	CD28/CD3ζ	Breast, ovarian, and renal cell carcinoma	Retrovirus	[214]
PBMC-NK	NKG2D ligands	CD3ζ with DAP10	Osteosarcoma, rhabdomyosarcoma, prostate carcinoma, colon carcinoma, gastric carcinoma, lung squamous cell carcinoma, hepatocellular carcinoma, and breast carcinoma	Retrovirus Electroporation (mRNA)	[215]
NK-92 cell line	HER-2	CD3ζ	Breast and ovarian carcinoma	Retrovirus	[216]
NK-92 cell line	HER-2	CD28/CD3ζ	Breast and pulmonary metastasis in a renal cell carcinoma model	Lentivirus	[217]
NK-92 cell line	EpCAM	CD28/CD3ζ	Breast carcinoma	Lentivirus	[218]
YT cell line	CEA	CD3ζ	Colon carcinoma	Electroporation (Plasmid DNA)	[219]
YTS cell line	PSCA	DAP12/CD3ζ	Prostate cancer	Lentivirus	[213]

**Table 3 cancers-16-02270-t003:** Examples of clinical trials with CAR-NK cells for the treatment of solid tumor.

Target Antigen	NK Cell Source	Targeted Disease	Trial Phase	Status	NCT Number
MUC1	Placental HSC-derived	Solid tumors (colorectal, gastric, pancreatic, NSCLC, breast, and glioma)	I/II	Unknown	NCT02839954
ROBO1	NK-92 cell line	Pancreatic cancer	I/II	Unknown	NCT03941457
ROBO1	Human primary NK cells	Solid tumors	I/II	Unknown	NCT03940820
NKG2D	Patient derived or donor NK cells	Metastatic solid tumors (e.g., colorectal cancer)	I	Unknown	NCT03415100
5T4 oncofoetal trophoblast glycoprotein (5T4)	Undisclosed	Advanced solid tumors	I	Unknown	NCT05194709
Claudin6, GPC3, Mesothelin, or AXL	Human primary NK cells	Advanced solid tumors (ovarian cancer and others)	I	Recruiting	NCT05410717
PD-L1	haNK	Solid tumors	I	Active, not recruiting	NCT04050709
PD-L1	haNK	Pancreatic cancer	I/II	Recruiting	NCT04390399

**Table 4 cancers-16-02270-t004:** Comparison of CAR-T, CAR-NK, and CAR-M immune cells (according to [250], partially modified in some positions for CAR-NK cells).

Factor	CAR-T Cells	CAR-NK Cells	CAR-M Cells
Source	Autologous or MHC-I matched allogeneic	Autologous or allogeneic; can be generated from different sources	Autologous or allogeneic; can be generated from different sources.
In vitro expansion	Yes	Yes	In autologous: OK In iPSCs and cell lines: required to expand before transduction
CAR transduction efficacy	Higher	Low	Low
Cytokines are used for cell expansion	IL-2	IL-15, IL-2, IL-21	GM-CSF
In vivo controlling of proliferation and expansion	Needed	Easier or not needed	Probably needed
Repeat activation upon first antigen exposure	Slow	Fast	Fast
Life span and persistence	High life span and long-term persistence	Low life span and limited persistency	Increased life span with limited persistency in circulation
Repeat doses	Only single dose	It is possible to treat patients with multiple doses	Only single dose
Tumor infiltration	Usually, poor	Usually, poor	Very abundant
Cytotoxicity effect	High	High	High
Cost	High	Low	Low
Off the shelf	Not significantly	Significantly	Possible with a different source of macrophage
Efficacy in solid tumors	Low	Moderate	High
Side effect	Common and often with fatality	Less common and low risk	It is expected to be shared without clinical evidence with low fatality potential

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
