# Peer review of "Contemporary Approaches to Immunotherapy of Solid Tumors"

_cancers, 2024, doi:10.3390/cancers16122270_

Round 1

Reviewer 1 Report

Comments and Suggestions for Authors

This review article discusses recent innovations in immunotherapy aimed at improving clinical efficacy in solid tumors, as well as strategies to overcome the limitations of various immunotherapies. It is well described nice review paper on important subjects in immunotherapy. From this basic positive viewpoint, I may recommend publication of this review article in Cancers. However, this review article has one fatal problem. Visual information such as figures and schemes are very less. This paper construction is totally bad for understanding by non-specialist readers. Therefore, I recommend the authors to add sufficient numbers of figures.

1) Please add initial figure to guide contents and story flow of this review article. This figure has to appear at the early stage of this review article. Appearance of such guiding figure would be a great help of understanding.

2) Similarly, please add one ending figure to explain general conclusion of this review articles. Even without dep understanding of the main points, easy understanding conclusive figures will convince many readers.

3) Hopefully, at least one figure had better be supplied to one section. The contents in this review article are basically rather complicated. Therefore, supports by visual items (figures and schemes) are crucial.

Author Response

First of all, we are very grateful to the reviewer for the attentive attitude to our paper. Thank you for the specified shortcomings, recommendations for changes, additions, and corrections. All changes in the manuscript are highlighted in red.

Visual information such as figures and schemes are very less. This paper construction is totally bad for understanding by non-specialist readers. Therefore, I recommend the authors to add sufficient numbers of figures. Please add initial figure to guide contents and story flow of this review article. This figure has to appear at the early stage of this review article. Appearance of such guiding figure would be a great help of understanding.

Agreeing with the reviewer's opinion, we have added 6 additional Figures that explain the immunotherapeutic approaches presented in the review: Figure 1. Schematic illustration of immune surveillance of the tumor microenvironment. (Lines 80-89); Figure 3. Targeting and mechanisms of ADC action. A detailed description is given in the text (Lines 256-257); Figure 4. Schematic illustration of the bispecific immune cell engagers that redirect immune cells such as T cells, natural killer (NK) cells, and cytotoxic/phagocytic cells against solid tumor cells (Lines 393-397); Figure 5. Structural details of chimeric antigen receptor (CAR)-T therapy (Lines 569-572); Figure 6. Schematic illustration of killing mechanisms of CAR-T (a), CAR-NK (b) and CAR-M (c) cells (Lines 597-598); Figure 7. Different strategies for immunotherapy of solid tumors (Lines 1217-1219).

Similarly, please add one ending figure to explain general conclusion of this review articles. Even without dep understanding of the main points, easy understanding conclusive figures will convince many readers.

We have added Figure 7 which summarises the data presented in the review. Figure 7. Different strategies for immunotherapy of solid tumors (Lines 1217-1219).

Hopefully, at least one figure had better be supplied to one section. The contents in this review article are basically rather complicated. Therefore, supports by visual items (figures and schemes) are crucial.

We hope that these changes will improve the quality of the manuscript.

Reviewer 2 Report

Comments and Suggestions for Authors

This is an amazing review. It is an exhaustive, updated and critical review about the current  approaches to immunotherapy of solid tumours. Authors show a high knowledge about this state-of-the art, the most recent advances, the most important clinical trials and so many details and relevant groups of research of the topics. The review is well referenced. According to the title, the review is focused on challenges and promising strategies for immunotherapy of solid tumours, although immunological treatments and  non-solid tumours are also comparative discussed. Section about CAR-T therapies, advances for the first to the fifth generation and limitations are brilliant. Conclusion introduces a simple “3C rule” that have sense and hopefully will be beneficial for immunotherapeutic approaches soon.

Only a couple of minor points to be addressed:

The review contains few schemes and illustrations. Only one figure and 4 Tables. About Figure, the classification is simplified, and only two subtypes are classified, cold and hot tumours. Concerning the legend of the Figure, I recommend that some comment about hot tumours would be added to fulfil the comparison between both types. For instance, the origin and role of dying cancer cells. As such, the legend is partial, and the comparison would be improved.

Table 4 is interesting summary for comparison of CAR-T, CAR-NK and CAR-M immune cells. The heading says that the Table is a modification of some Table/Data published at reference [249].  The modifications should be briefly discussed.

Author Response

We express our deep gratitude to the reviewer for the hard work of reading the paper and for all the comments and suggestions made for necessary corrections. Below is the list of all the questions and our answers to them. All changes in the manuscript are highlighted in red.

The review contains few schemes and illustrations. Only one figure and 4 Tables. About Figure, the classification is simplified, and only two subtypes are classified, cold and hot tumours. Concerning the legend of the Figure, I recommend that some comment about hot tumours would be added to fulfil the comparison between both types. For instance, the origin and role of dying cancer cells. As such, the legend is partial, and the comparison would be improved.

Agreeing with the reviewer's opinion, we have added 6 additional Figures that explain the immunotherapeutic approaches presented in the review. Figure 1. Schematic illustration of immune surveillance of the tumor microenvironment. (Lines 80-89); Figure 3. Targeting and mechanisms of ADC action. A detailed description is given in the text (Lines 256-257); Figure 4. Schematic illustration of the bispecific immune cell engagers that redirect im-mune cells such as T cells, natural killer (NK) cells, and cytotoxic/phagocytic cells against solid tumor cells (Lines 393-397); Figure 5. Structural details of chimeric antigen receptor (CAR)-T therapy (Lines 569-572); Figure 6. Schematic illustration of killing mechanisms of CAR-T (a), CAR-NK (b) and CAR-M (c) cells (Lines 597-598); Figure 7. Different strategies for immunotherapy of solid tumors (Lines 1217-1219). We have corrected the legend and the Figure 2 (formerly Figure 1). Lines 184-194

Table 4 is interesting summary for comparison of CAR-T, CAR-NK and CAR-M immune cells. The heading says that the Table is a modification of some Table/Data published at reference [249].  The modifications should be briefly discussed.

We have added an additional explanation to Table 4 to reflect new CAR-NK data reflected in the review (Line 1209).

Additionally, we added an important, in our opinion, information regarding the fifth generation CAR-T (Lines 587-592).

We hope that these changes will improve the quality of the manuscript.